# The polarizing impact of numeracy, economic literacy, and science literacy on the perception of immigration

**Lucia Savadori** [1]*, **Maria Michela Dickson**[1], **Rocco Micciolo**[2], **Giuseppe Espa**[1]

**1** Department of Economics and Management, University of Trento, Trento, Italy, **2** Department of Psychology and Cognitive Sciences, University of Trento, Trento, Italy

* lucia.savadori@unitn.it

**Data Availability Statement:** Data are available at https://osf.io/mzt85/ for transparency, but no private use is allowed since they are property of the Trento Municipality.

## Abstract

Immigrants might be perceived as a threat to a country's jobs, security, and cultural identity. In this study, we aimed to test whether individuals with higher numerical, scientific, and economic literacy were more polarized in their perception of immigration, depending on their cultural worldview orientation. We measured these variables in a representative sample of citizens in a medium-sized city in northern Italy. We found evidence that numerical, scientific, and economic literacy polarize concerns about immigration aligning them to people's worldview orientations. Individuals with higher numerical, economic, and scientific literacy were less concerned about immigration if they held an egalitarian-communitarian worldview, while they were more concerned about immigration if they held a hierarchical-individualistic worldview. On the contrary, individuals with less numerical, economic, and scientific literacy did not show a polarized perception of immigration. Results reveal that citizens with higher knowledge and ability presented a more polarized perception of immigration. Conclusions highlight the central role of cultural worldviews over information theories in shaping concerns about immigration.

## 1. Introduction

Immigration has been steadily rising in all European countries until the mid-1990s [1], but the strong wave of asylum-seeking migrants in recent years has caused protests and tensions. More than a third of Europeans consider immigration the most important issue facing the European Union [2]. Moreover, many Western European countries (Germany, France, the UK, and Italy) have lately experienced a significant increase in anti-immigration parties' support [3].

Immigrants might be perceived as a threat to jobs, culture, and the security of a country [4]. According to the cultural theory of risk, people perceive something as risky when their cultural worldview is challenged [5]. Dake [6] describes worldviews as general attitudes that people have towards the world and its social organization. The role that worldviews play in determining risk perception has recently received new attention thanks to the cultural cognition theory

**Funding:** All the authors are thankful to the Istituto di Scienze Della Sicurezza (ISSTN; Institute of Security Sciences; https://projects.unitn.it/isstn/) at the University of Trento and the Comune di Trento (Trento Municipality) for logistic support in data collection. The ISSTN and Comune di Trento provided material support for collecting the data for this study. Specifically, they were responsible for sending out, by regular mail, letters inviting respondents to fill out the online questionnaire and providing telephone assistance to those respondents who required help filling out the online questionnaire. The funders had no role in study design, data analysis, decision to publish, or preparation of the manuscript. The authors and the study received no financial support from the founders. None of the authors received a salary from the founders.

**Competing interests:** The authors have declared that no competing interests exist.

[7, 8]. Cultural cognition theory defines four cultural values grouped along two dimensions: hierarchy-egalitarianism (hierarchy) and individualism-communitarianism (individualism) [9, 10]. Individuals with a hierarchical worldview prefer a hierarchical society clustered according to well-defined differences among groups identified by gender, race, and class (e.g., "It seems like blacks, women, homosexuals, and other groups don't want equal rights, they want special rights just for them"). On the opposite side, people with an egalitarian worldview prefer a society where minorities have equal rights and inequalities are eradicated (e.g., "We need to dramatically reduce inequalities between the rich and the poor, whites and people of color, and men and women"). People with an individualist worldview prefer a society where individuals will secure their well-being without assistance or interference from society (e.g., "The government interferes far too much in our everyday lives"). On the opposite pole, people who hold a communitarian worldview think that the government has the responsibility for collective welfare and the power to override individual interests (e.g., "Sometimes the government needs to make laws that keep people from hurting themselves").

Cultural worldviews have been shown to explain the risk judgments of individuals in the face of threats [6, 8, 11–21]. For example, egalitarian individuals perceive higher risk and tend to be more concerned about nuclear power plants than hierarchical individuals [22]. Individuals with hierarchical and individualistic worldviews are also less concerned about climate change [10, 15, 16] and are less willing to change their behaviors and say they support policies to constrain climate change [15]. In general, a growing body of evidence shows that egalitarian and communitarian individuals have a higher perception of environmental and health risks, whereas hierarchical and individualistic individuals perceive fewer environmental and health risks [17–21].

To the extent that citizens view immigration as a potential threat, we expect that cultural worldviews would explain individual differences in the perception of immigration. There are no studies yet that relate hierarchical and individualistic orientations to perceptions of immigration, but there are some data that point in this direction. For example, the ideological attitude of right-wing authoritarianism was a robust predictor of anti-immigrant attitudes in countries where immigrants were perceived as increasing the crime rate and not being beneficial to the economy (e.g., Germany, Italy) [23]. Right-wing authoritarianism is a combination of three attitudes that are very similar to the value of hierarchy: conventionalism (agreement with traditional societal norms), authoritarian submission (tendency to obey authority figures who represent these norms), and authoritarian aggression (willingness to engage in authority-sanctioned aggression toward individuals or groups that violate traditional norms) [24].

In the present study, we measured both worldviews of hierarchy–egalitarianism and individualism–communitarianism, under the assumption that both hierarchical and individualistic worldviews would be associated with increased concern about immigration compared to egalitarian and communitarian worldviews.

## 2. State of the art

In recent years, polarized perceptions of the world have been found to be exacerbated by education and knowledge [10, 25, 26]. For example, an extensive survey conducted in the United States by Kahan and colleagues [10] found that scientific literacy was only weakly negatively correlated with concern about climate change, whereas hierarchical and egalitarian worldviews primarily moderated this relationship. From these data, the authors concluded that communication efforts about real or presumed threats should not focus on conveying objective knowledge because it would neither increase nor decrease public concern about threats [10]. Moreover, the same data would seem to indicate a counterintuitive role of knowledge.

Individuals who showed more objective knowledge were also more polarized in their opinions, depending on their cultural or political orientations [10].

More in general, people's political orientation has been shown to polarize public opinion of more educated individuals on controversial issues such as climate change or stem cell research. The polarization effect has a typical funnel pattern [25] since the gap between the beliefs of political conservatives and political liberals widens as the level of education increases, mimicking a funnel shape. The more educated and scientifically literate individuals sustain the need to contrast presumed threats more if they are left-oriented than if they are right-oriented. In contrast, the less educated and lower scientifically literate individuals do not show such a diverging trend in their opinions [10, 25, 27–39]. These studies raised questions about which science communication model was best suited to explain them. On the one hand, the "deficit" model predicts that greater dissemination of scientific knowledge will increase public consensus toward scientific standpoints (e.g., reduce climate change) [40]. On the other hand, the evidence shows the opposite, namely that greater knowledge increases the polarization of public opinion toward opposite poles defined by pre-held ideological orientations. A "motivated reasoning" model has been suggested to explain this evidence, suggesting that people filter and process information to support previously held beliefs [41].

Climate change and environmental issues are not the only controversial topics for which the funnel pattern has been observed. The same pattern was also detected in opinion about vaccines [42, 43], support for embryonic stem cell research [25, 44], and opinion toward the Big Bang and human evolution [25], as well as COVID-19 risk perception [30]. Inconsistencies were also found. There was, for example, no interaction between knowledge and political or religious identity on nanotechnology and genetically modified food. The authors advanced the explanation that these issues generated controversy, but they did not become part of the most significant social conflicts in America [25]. Seemingly, numeracy and ideology interacted to predict COVID-19 risk perception but failed to do so when verbal ability was also measured, suggesting the possibility that different predictors might drive greater polarization depending on the issue examined [30].

Most studies on the polarization of beliefs used political orientation as the polarizing variable. Instead, following Kahan et al. [10], we used a more nuanced underlying ideological measure, i.e., worldviews, to capture where the individual lies on the ideological spectrum represented by hierarchical-individualistic views on one side and egalitarian-communitarian views on the other. Individual preferences on social problems such as gun control, nuclear waste disposals, COVID-19, and climate change are strongly influenced by cultural worldviews [9, 11, 12, 45]. Indeed, in a study of 6,991 individuals across the world, an individualistic worldview predicted COVID-19-related attitudes and behaviors more than all other variables (including political orientation) in five out of the ten countries surveyed (UK, Germany, Sweden, Spain, and Japan) [11]. Moreover, worldviews have been successfully used in a prior study on risk perception and the polarizing impact of knowledge [10]. We, therefore, measured individual worldviews to categorize individuals into hierarchical-individualistic vs. egalitarian-communitarian and measure the polarizing impact that this underlying ideology induces when put in interaction with personal knowledge.

Past studies on the polarizing effect of knowledge have mainly focused on environmental or medical topics such as climate change or COVID-19, ignoring other societal issues, such as immigration. We suggest that to the extent that society is becoming more culturally complex, people's knowledge might also become critical for judging and making decisions on immigration issues. However, more knowledge might not be linearly related to more or less concern about immigration. More likely, knowledge might polarize perceptions of immigration according to people's cultural worldviews. Our study's main goal was to examine the impact of basic

economic and scientific knowledge and numerical ability on concerns about immigration when controlling for cultural worldviews. More precisely, we expected cultural worldviews to moderate the relationship between knowledge and public concern about immigration.

Of central importance to the knowledge-related polarization effect is the type of knowledge/education/ability considered in the interaction [30]. Across studies on polarization, researchers have mostly used education as the knowledge variable to test for the interaction between partnership and opinions, with few exceptions that used science knowledge and others that used cognitive abilities (i.e., numeracy and verbal abilities). The type of knowledge variable chosen has been shown to determine the chance of detecting the interaction with ideology [30]. When verbal ability measures were not controlled for, numeracy and ideology did interact to predict outcomes, but they failed to interact when the verbal ability measure was included in the model [30]. Thus, in the present study, we chose to examine the hypothesized interaction effect (knowledge x ideology), exploring multiple knowledge variables. We used two knowledge predictors (science and economic literacy) and one cognitive ability predictor (numeracy) to broaden the set of knowledge variables that elicit the knowledge-related polarization effect.

## 2.1 Economic literacy

Economic knowledge, also called economic literacy, is the personal knowledge about basic economic concepts, such as markets and prices, supply and demand, money and inflation, economic institutions, labor markets, and income [46]. Economic literacy must not be confused with financial literacy. Financial literacy is the knowledge about concepts of financial management, budget, and investment, such as risk diversification, interest compounding, mortgages, and other debt instruments, just to name a few [47].

Economic knowledge, opposition to immigration, and ideological orientation are closely interrelated variables. For example, perception of the economic threat posed by immigrants and opposition to immigrants are both associated with people's political preferences. A negative attitude toward immigrants has been shown to be the most important predictor in explaining support for far-right-wing parties [e.g., 48]. Seemingly, the perceived economic threat posed by immigrants has been shown to increase public preference for right-wing parties [e.g., 49]. Not surprisingly, objective economic knowledge was found to be related to ideological beliefs [50, 51]. Indeed, economic knowledge significantly influenced public opinions on governmental policies (e.g., "the U.S. government should prohibit the increase of oil and gas prices") [50], an issue closely related to the individualism-communitarianism worldview (e.g., "government interferes too much in our daily lives") [9]. Moreover, economic knowledge predicts individuals' political affiliation [51], a variable typically used in studies showing the knowledge-related polarization effect [e.g., 25]. Besides, economic knowledge is also associated with education [52], another central variable typically used in studies showing the knowledge-related polarization effect [e.g., 25]. Finally, economic issues have also been used as an outcome variable in knowledge-related polarization studies [31]. An earlier study showed that education interacted with political orientations to determine beliefs about economic growth. Individuals who were both more educated and right-wing oriented believed more strongly that economic growth created jobs, happiness, and public services, while those who were highly educated and left-wing oriented rejected this belief more strongly [31].

We, therefore, expected that economic knowledge would be an important variable in detecting the knowledge-related polarization effect when dealing with immigration issues. Our hypothesis was that perceptions of immigration would be moderated by economic knowledge such that greater knowledge would be associated with greater worldview polarization.

Ideological views on immigrants are indeed strongly divergent. On the one hand, the left-wing narrative argues that immigration can benefit the economy, providing companies with skilled workers, and relieving tension on the tax-funded pension system threatened by the lack of population growth. On the other hand, the right-wing rhetoric about immigrants centers on the fear that immigrants might take jobs away from local workers and take more from the government, in the form of social services, than they give back in taxes. In line with knowledge-related polarization studies, we predicted that economic knowledge would further exaggerate these already polarized beliefs. We hypothesized that individuals who were both highly knowledgeable in economic issues and egalitarian-communitarian oriented would be more favorable toward immigration, while those who were highly knowledgeable in economic issues and hierarchical-individualistic oriented would oppose immigration and immigrants more strongly.

## 2.2 Numeracy

Numeracy refers to individuals' capacity to comprehend and use quantitative information effectively [53]. There is evidence that individuals' numerical skills are relatively low [54, 55]. However, a growing body of research shows that numerical ability is critically important to the judgments and quality of decisions we make [56]. Individuals with higher numerical ability are more adept at deriving more precise affective meaning from numbers that will later be used to form their perceptions of risk and to make choices in the face of threats. Instead of using the affective meaning of numbers, individuals with less numerical ability use mental heuristics and rely on emotional reactions [57]. Less numerate individuals are not only less accurate, but they also overestimate risk more. For example, less numerate women overestimate their personal risk of breast cancer compared to highly numerate women while controlling for demographic characteristics [58]. Less numerate individuals were also more responsive to narratives than information on the objective likelihood, and they held a systematically higher risk perception [59]. Less numerate individuals were also more susceptible to motivated reasoning related to the Muslim immigration ban [60].

Some previous studies on knowledge-related polarization used individuals' numeracy to predict polarization [10, 30]. More numerate individuals were found to be more concerned with climate change risks if they held an egalitarian-communitarian worldview than if they held a hierarchical-individualistic worldview [10]. In a subsequent study, numeracy and ideology interacted to predict COVID-19 risk perceptions when another ability, i.e., verbal ability, was not controlled for, whereas the numeracy x ideology interaction did not predict outcomes when the other ability was included in the model [30]. The authors, therefore, suggest that verbal ability should always be measured and contrasted with numerical ability in the analyses. But they also suggest that the role of numerical ability might depend on the type of issue being examined. If the task requires numerical ability rather than verbal ability, then a greater polarization would be seen with numeracy and not verbal ability. As regards the specific issue object of the current study, i.e., immigration, we hypothesized that numeracy would be a significant predictor of knowledge-related polarization. More numerate individuals are generally better able to derive meaning from numbers [61]. However, more numerate individuals also form a less clear mental image of the people in need, and their willingness to help depends less on affective cues (i.e., mental images, presentation format) and more on extensional cues, such as the estimated impact of the donation [62] and the proportion of victims helped [63]. Thus, we anticipated that more numerate individuals might be more susceptible to motivated reasoning processes and show greater polarization as they would be more able to extract meaning from numerical information (e.g., immigration rates and crime rates) coherent with their ideological views. We, therefore, predicted that higher numerical skills would influence public opinion

on immigration, depending on the individual worldview orientation. Individuals with both a high numeracy level and an egalitarian-communitarian orientation would show a much more favorable view of immigration, while those with both a high numeracy level and a hierarchical-individualistic orientation would show a much less favorable view.

## 2.3 Science literacy

Science literacy is the knowledge of basic scientific facts [64]. While scientific knowledge is usually not a significant predictor of risk perception *per se* [10, 65], it is a significant factor in polarizing public opinions about climate change [10, 25]. Individuals with higher science literacy showed the greatest cultural-worldviews polarization for climate change risks [10]. Seemingly, individuals with greater science knowledge showed more political polarization on issues such as stem cell research, the big bang, human evolution, and climate change [25]. On a similar line, greater attention to scientific news increased support for policies aimed at reducing climate change for strong liberals but reduced support for strong conservatives [66]. Science knowledge and fear of immigration share a common ground when it comes to viruses and diseases. Indeed, people might fear immigrants thinking that they can be vehicles for viruses and diseases. Concerns about immigrants and disease have been constantly registered throughout history. For example, Markel and Stern [67] explored why in three periods, from 1880 to the present, immigrants have been stigmatized as the etiology of a variety of diseases, despite the data do not support such a narrative. Human mobility, indeed, was historically associated with the spread of infectious diseases [68], however this relationship no longer exists in the contemporary age. Despite this, fear is still supported by the media who portray immigrants as disease spreaders [69]. For example, during the of COVID-19 outbreak in Italy, the question of whether more immigrants should be brought into the country from the sea borders was also deeply intertwined with the threat they posed as positive drivers of viral infection [70]. However, this might be especially true for individuals opposing immigration who historically hold a more ideological attitude of right-wing authoritarianism. We, therefore, predicted that greater science knowledge would interact with cultural worldview orientations in explaining public opinion on immigration. Thus, we anticipated those people with greater science knowledge, and egalitarian-communitarian orientation might show more extreme positive opinions on immigration, while those people with greater science knowledge and hierarchical-individualistic orientation might show more strong negative opinions on immigration.

## 3. The PRI survey

The "PRI: Perception of Risks connected to Immigration" survey was carried out on a representative sample of inhabitants in Trento (Italy) from March 1 to April 30, 2019. The survey's main goal was to examine the impact of numerical, scientific, and economic literacy on immigration concerns when controlling for cultural worldviews. More precisely, we expected numerical ability, knowledge about the economy, and knowledge about basic science facts to positively influence public opinion of immigration among individuals holding an egalitarian-communitarian worldview orientation but negatively influence public opinion of immigration among individuals sharing a hierarchical-individualistic worldview.

The reference population of the survey was selected from the municipal register updated on January 1, 2019. The reference population included all adult citizens between 18 and 80 years of age of both sexes, and residents in the city's territory, for a total of 90,051 units. Homeless and nomads were excluded from the reference population.

A stratified sampling design was adopted: the population was divided into homogeneous, non-overlapping strata (groups) based on known stratification variables available for all units.

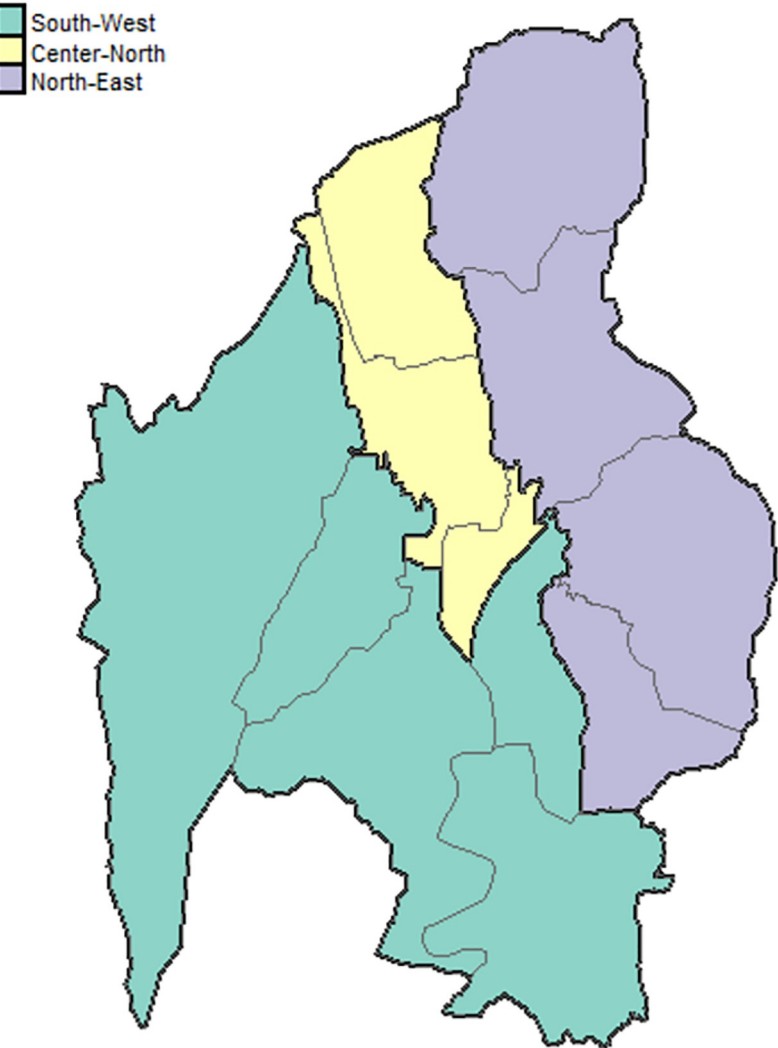

**Fig 1. Map of the city of Trento (Italy).** Black contours identify the 12 districts, and the three macro-areas are identified by three colors. The yellow area corresponds to the old town of the city, and it is the most populated zone.

The variables adopted for the stratification were the following: gender (female, male), date of birth (from 18 to 35 years, from 36 to 55 years, and from 56 to 80 years), and the district of residence. Regarding the latter, the city of Trento is divided into 12 districts of residence, which in the present study were grouped into three macro-areas (Fig 1), according to demographic and social features of thereof: South-West, Centre-North, North-East.

The population resulted in a multivariate stratification with a total of 18 strata (see S1 Table). The allocation of the sample in each stratum was proportional to the stratum size, with a minimum number of units in each stratification cell $n_h = 5$ and a maximum $n_h = N_h$. When $n_h < 5$, the allocation is forced to $n_h = 5$ and when $n_h = 5$, the stratum is censused. The stratified sample has been selected by using the R package `sampling` [71]. The selection criterion in each stratum has been the simple random sampling without replacement, with inclusion probability for the $h-th$ stratum equal to $\pi_h = \frac{n_h}{N_h}$. The obtained sample was composed of 2,008 adult inhabitants, with a sampling fraction equal to 2%. This is a notable issue since, in

national social surveys, the sampling fraction rarely exceeds 0.1% (see S2 Table for the allocation of the selected sample).

Participation was voluntary. In total, 551 people from the city of Trento completed the survey. The overall response rate was 27% (551 questionnaires completed out of 2,008), a remarkable result compared to opinion surveys conducted in similar ways (see S3 Table for the sample response rate according to the stratification variables).

The survey was conducted by means of a questionnaire administered with CAWI/CATI methodologies through the statistical survey web app LimeSurvey. Units were contacted by letter and invited to autonomously participate in the online survey or to contact by telephone the research group for help in compiling the survey. The data collection was approved by the Research Ethics Committee of the University of Trento. Each participant was sent a unique (and anonymous) code identifier by mail with which they could access the online questionnaire. A single usage of the code was allowed. No signed informed consent was collected, but the participant gave electronic informed consent by accessing the questionnaire with their unique code and agreeing to complete it online. The participant needing help in compiling the form contacted the telephone number and provided their unique code to the experimenter, who accessed the questionnaire on their behalf and read the questions to the participant by phone and completed the questionnaire.

The set of respondents included 281 males (51%) and 270 females (49%) with an average age of 52.2 years (SD = 16.4). More than a third of the respondents (38.3%) had completed at least a Bachelor's degree, whereas 44.1% had completed at least a high-school diploma, and the remaining (17.6%) had completed lower levels of education (or had no certificates).

Before proceeding with the analysis, we corrected the data for non-responses [72] to produce unbiased estimates. To this aim, a calibration estimator was implemented [73], forcing the calibration on the three stratification variables totals. The implementation was carried out through the R package survey [74].

## 3.1 The questionnaire

The questionnaire was composed of several sections. At the beginning of each section, there was a short introduction. In the section dedicated to the measurement of perception of immigration, instructions read: "There are different opinions on immigrants living in Italy. By "immigrants" we mean people from other countries who come to settle in Italy. Here below we will ask you some questions about your opinions". In the section dedicated to measuring the cultural worldviews, instructions read: "In this section, we will ask you a series of questions about your socio-economic opinions. Please indicate your degree of agreement or disagreement with each statement". In the sections dedicated to the measurement of numeracy, economic literacy, and science literacy, it was repeated each time: "The questions in this section serve to measure your familiarity with [mathematics and probability; economic phenomena; scientific issues]. The questions do not in any way measure intelligence but rather [the habit you have formed over time to use your knowledge to perform mathematical and probabilistic calculations; the knowledge formed over time through study, work, personal interests, and media]. You may not feel confident in answering some questions, but we invite you to find the answer that seems to be more correct". At the end of the questionnaire, participants answered some demographic questions on gender, age, place of residence, and education. The questionnaire was in Italian, the respondents' native language.

**3.1.1 Cultural worldviews.** To measure cultural worldviews, we used the short version of the cultural worldviews scale proposed by Kahan et al. [9]. The scale consisted of 12 statements, and the participant was asked to report the degree of agreement with each of them (see

S4 and S5 Tables for items and descriptive statistics). The statements have been designed to reflect two underlying bipolar dimensions (hierarchy vs. egalitarianism and individualism vs. communitarianism). For all items, participants indicated the degree of agreement or disagreement on a five-point scale (1 = completely agree; 5 = completely disagree). We gave the option of answering: "I do not know / I do not answer." Answers were treated as numeric and then averaged in such a way that higher scores correspond to a socio-politic orientation attributable to a conservative profile (hierarchical-individualist). A reliable composite overall index (α = .73) expressing the degree of hierarchical-individualist worldview was obtained by averaging all the responses (M = 2.63; SD = 0.45). This scale was successfully used in European populations such as the UK [75], Dutch [76, 77], and Swiss populations [15].

**3.1.2 Perception of immigration.** Perception of immigration was measured by asking participants to express their agreement with ten statements derived from the General Social Survey 1972–2014 [78] and answer three questions about risk perception (see S6 and S7 Tables for items and descriptive statistics). The ten statements from the General Social Survey asked about the perception of several aspects of immigration. Participants answered on a five-point scale (1 = completely agree; 5 = completely disagree). The risk perception questions asked about the risks and benefits of immigration for Italian society as a whole [79, 80] and the extent to which thinking about immigration was associated with positive or negative emotions as a measure of affective attitude, a component of risk perception [81–83]. Participants answered on a five-point scale, both the risk and benefit perception questions (1 = not at all risky/beneficial; 5 = extremely risky/beneficial) and the emotion question (1 = very negative; 5 = very positive). We gave the option of answering: "I do not know / I do not answer." All items were reverse coded so that higher values corresponded to a more favorable attitude toward immigration. Responses were grouped in a reliable composite average index (α = .94; M = 3.16; SD = 0.75) reflecting the degree of tolerance toward immigration. To be conservative, the 13 items were subjected to a principal component analysis (PCA) which showed that the principal component explains about 60% of the total variability and that the weights of the items (the factor loadings of the PCA) are substantially equal to each other. In addition, the arithmetic mean of the 13 items has a very high correlation index (r = .98) with the score given by the first principal component (the scores, i.e., the values of the new variable defined, precisely, by the first principal component). We, therefore, considered it more informative to use the simple arithmetic mean as the dependent variable and not a more complex statistic, such as the mean of the items each weighted by the principal component, also because the arithmetic mean entails a lesser loss of information due to missing cases.

**3.1.3 Numeracy.** Five questions from Weller et al. [84] were used to measure objective numeracy (see S8 and S9 Tables for items and descriptive statistics). Each question had four possible answers, but only one was correct. To compute the numeracy index, we calculated the number of correct answers (M = 3.22; SD = 1.11). Some of the items had to be adapted to the cultural context. We gave the option of answering: "I do not know / I do not answer".

**3.1.4 Economic literacy.** We used 12 questions from the Test of Economic Knowledge [46] to measure economic literacy. The items n.8, 9, 13, 15, 17, 23, 25, 26, 30, 41, 42, and 44 of the original test by Walstad et al. [46] were used (see S10 and S11 Tables for items and descriptive statistics). Items were selected to meet two criteria: (1) they had to be sufficiently easy (more than 40% correct responses in the U.S. student sample not enrolled in a basic course with economics, as reported in [46]); (2) they had to be representative of different test contents. The subset of 12 items selected had a difficulty that ranged from a minimum of 42.3% to a maximum of 66.3% correct responses, with an average of 51% correct responses. In addition, the items investigated the following 9 contents out of 20: (a) Economic incentives—prices, wages, profits, etc. (item 8); (b) Voluntary exchange and trade (item 9); (c) Markets and prices

(item 13); (d) Supply and demand (items 15 and 17); (e) Money and inflation (items 23 and 25); (f) Interest rates (item 26); (g) Entrepreneurship (item 30); (h) Unemployment and inflation (items 41 and item 42); (i) Fiscal and monetary policy (item 44). Individual score on economic literacy was computed by calculating the number of correct answers (M = 8.91; SD = 2.67). We gave the option of answering: "I do not know / I do not answer."

**3.1.5 Science literacy.**  The questions to measure scientific knowledge were drawn from the Science & Engineering Indicators [85] (see S12 and S13 Tables for items and descriptive statistics). They are ten statements to which participants replied "true" or "false." The statements span from clinical aspects (e.g., antibiotics kill viruses, as well as bacteria) to biological aspects (e.g., it is the paternal gene that determines whether the child will be a male or a female) but also technological aspects (e.g., lasers work by focusing sound waves). The scientific knowledge index was calculated, considering the number of correct answers (M = 8.39; SD = 1.70). We gave the option of answering: "I do not know / I do not answer."

# 4. Results of the study

## 4.1 Analytical strategy

To test whether cultural polarization was greater among respondents with higher knowledge, we fit a model predicting participants' perception of immigration (coded such that higher values represent a more positive attitude) as a function of measures of numeracy, economic literacy, science literacy, and a new compound measure of *total literacy* representing the mean of the aggregated knowledge measures. In each model, we estimated the direct effect of each explanatory variable on the perception of immigration as well as one interaction term—each interaction term combined worldview orientation with one of the four knowledge measures.

In the models, the response variable (perception of immigration) was computed for each individual *i*, as the mean of the scores of the ten statements from the General Social Survey and the three questions about risk perception. Hence, the estimated model in all four measures is $\hat{y}_i = b_0 + b_1 x_i + b_2 z_i + b_3 x_i + z_i$, where $z$ is a dichotomous variable created to label individuals as egalitarian-communitarians or hierarchical-individualists, and the explanatory variable $x$ represents, in the order in which the models are presented, respectively numeracy, economic literacy, science literacy, and total literacy. To create the $z$ variable, we averaged individual responses to the 12 statements of the cultural worldviews scale (say $w$) such that higher scores correspond to a higher hierarchical-individualistic orientation. The cut-off was fixed at the median level of 2.5, such that:

$$z_i = \{ \begin{array}{l} 1 \; if \; \bar{w}_i > 2.5 \; (\text{hierarchical} - \text{individualist}) \\ 0 \; if \; \bar{w}_i < 2.5 \; (\text{egalitarian} - \text{communitarian}). \end{array} \tag{1}$$

This dichotomization does not constitute a forcing nor an excessive simplification and, especially, does not introduce any distortion in the proposed and estimated models, such as significant interactions which would otherwise not exist. Dichotomization can yield misleading results in the presence of continuous reality [86]. However, we believe that our reality is made up of two groups with a certain degree of overlap. The cultural worldviews scale is conceived to identify them, of course, with the possibility of making classification errors. The worldview orientation scale uses a continuous measurement for research purposes, i.e., the need to elicit truthful answers to sensitive ideological questions. However, the items measuring cultural worldviews are aimed at producing a dichotomization, i.e., a classification of an individual as hierarchical-individualistic or egalitarian-communitarian. Following MacCallum et al. [86], we performed a simulation that demonstrates how using the continuous variable to

**Table 1. Linear regressions results.**

| | Explanatory literacy variables | | | |
|---|---|---|---|---|
| | **Numeracy** | **Economic literacy** | **Science literacy** | **Total literacy** |
| Intercept ($b_0$) | 3.15*** (0.18) | 3.00*** (0.18) | 2.75*** (0.29) | 2.59*** (0.27) |
| Literacy ($b_1$) | 0.12* (0.05) | 0.06** (0.02) | 0.09** (0.03) | 0.05*** (0.01) |
| Worldviews ($b_2$) | -0.19 (0.20) | -0.20 (0.22) | 0.03 (0.34) | 0.21 (0.31) |
| Literacy x worldviews ($b_3$) | -0.14* (0.059) | -0.05* (0.02) | -0.08* (0.04) | -0.04** (0.01) |
| RSE | 0.67 | 0.67 | 0.67 | 0.66 |
| $R^2$ | 0.20 | 0.20 | 0.20 | 0.21 |
| Adjusted $R^2$ | 0.20 | 0.20 | 0.19 | 0.20 |
| F | 54.56*** | 47.43*** | 46.38*** | 48.75*** |
| AIC | 1120.76 | 1116.23 | 1118.77 | 1113.11 |

Linear regressions results predict perception of immigration from the explanatory literacy variables (numeracy, economic literacy, science literacy, and total literacy), worldviews (egalitarian-communitarians vs. hierarchical-individualists), and the interaction between them.

*Note*. SEs are shown in parentheses. RSE = Residual standard error. AIC = Akaike information criterion;

***p-value < 0.000,

**p-value < 0.001,

*p-value < 0.050.

predict a dichotomized reality reduces the power of the interaction test (which is the parameter of our primary interest). We show some of the most relevant results that emerged from that simulation in the attached file (see S1 File).

Using the threshold at the median level, we had 214 cases for $z = 0$, and 334 cases for $z = 1$. Therefore, the estimated prediction equation assumes then the form $\hat{y}_i = b_0 + b_1 x_i$, if $z = 0$, and $\hat{y}_i = (b_0 + b_2) + (b_1 + b_3)x_i$, if $z = 1$. Moreover, each $x$ is the sum of the correct answers given by each individual to the questions of each knowledge test. Using the sum is an ideal strategy in the presence of "I don't know/I don't answer" as a possible response.

Table 1 shows the results of the multiple regression models for the four considered explanatory variables.

## 4.2 Main effects

Examining Table 1, it is worth noting that all the parameter estimates of the literacy variables ($b_1$) have a positive sign denoting a positive association between the perception of immigration and each of the knowledge measures: numeracy, economic literacy, science literacy, and total literacy. The direct effects on the perception of immigration predicted by the model, by controlling for the other predictors, are stronger for economic and science literacy than numeracy. In all cases, however, the associations are significant (all *p-values* < .05). Participants' perception of immigration is significantly predicted by their scores on the numeracy test, the science literacy test, and the economic literacy test. Participants scoring higher on these measures showed a more positive perception of immigration.

The situation is different for worldviews (the dummy variable) for which the regression coefficients of the direct effects are not significant (see $b_2$-row). The issue does not deserve special attention because the variable worldviews is contained in the interaction term that assumes, on the contrary, crucial importance, as discussed hereafter. It is sufficient to note that a non-significant regression coefficient for worldviews means that an individual classified as hierarchical-individualist, with no numerical, economic, scientific, and total literacy skills, has, on average, the same perception of immigration as another individual classified as egalitarian-

communitarian. In other terms, the lack of these skills makes the contribution of worldviews indifferent to the model predicting perception of immigration.

## 4.3 Interactions with worldviews

The $b_3$-row in Table 1 reports the estimates of the regression coefficients for the interaction among worldviews and each explanatory literacy variable used in the four distinct models. All the coefficients have negative signs, and all interaction terms are significant. The reverse association is strongly significant (*p-value* < 1‰) for total literacy. However, it is also significant (*p-value* < 5%) for numeracy, economic literacy, and science literacy. Therefore, we do not eliminate, even if not significant, the main effect of worldviews in all four estimated models because, as said before, it appears in the significant interactions. Moreover, the slope of the relationship between the perception of immigration and each literacy explanatory variable changes when we go from the egalitarian-communitarian worldview to the hierarchical-individualistic one. It means that the two lines are not parallel, as shown in Fig 2.

Individuals classified as egalitarian-communitarian increase their positive perception of immigration as their skills in numbers and their knowledge in economics and science, increase. On the contrary, individuals classified as hierarchical individualists do not increase their positive perception of immigration as their competence in these domains increases, showing a uniform perception of immigration. Roughly speaking, hierarchical-individualistic individuals show a strong homogeneity in their perceptions of immigration, while egalitarian-communitarian individuals demonstrate a relevant heterogeneity in their perceptions of immigration, as explained by the four knowledge measures.

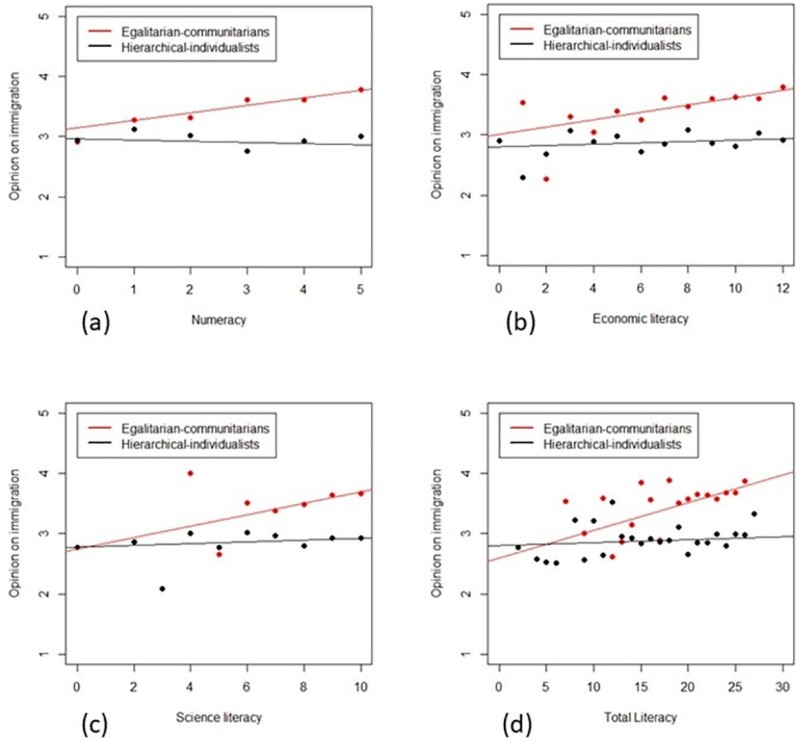

**Fig 2. Interactions.** Interaction among worldviews with, respectively, numeracy (a), economic literacy (b), science literacy (c), and total literacy (d). Red and black lines represent the fitted values for, respectively, egalitarian-communitarian and hierarchical-individualistic individuals. Red and black dots represent the mean values of perception of immigration for, respectively, egalitarian-communitarian and hierarchical-individualistic individuals.

The last row of Table 1 shows the (Akaike information criterion; AIC) values for the four fitted models. The model with the lowest AIC included total literacy as the explanatory variable, while the model with the highest AIC included numeracy as the explanatory variable. The difference between these two AICs was 7.651, indicating, according to Burnham and Anderson [87], good support for considering the model that included total literacy as an explanatory variable as a better model. A similar result, even if with weaker evidence, was found when considering the AIC difference between this model and those which included economic literacy (3.137) or science literacy (5.658) as the explanatory variable.

## 5. Discussion and conclusions

Previous studies found that polarized perceptions of controversial issues, such as climate change, can be exacerbated by education and scientific knowledge [e.g., 25]. Here, we examined how a similar pattern emerges for the perception of immigration in a representative sample of citizens of a southern mid-sized European city. Overall, we confirmed that cultural polarization, measured by cultural worldviews, was greater among individuals with higher knowledge, measured by numerical, economic, and scientific literacy tests. We have observed that a hierarchical-individualistic worldview was significantly associated with more concern about immigration than an egalitarian-communitarian worldview, and this gap was more remarkable for individuals with higher numeracy, greater economic literacy, and greater scientific literacy.

These results are consistent with previous research that has found a political polarization of beliefs on scientific issues [e.g., 10, 25]. But our study is novel in several respects. While previous studies have predominantly examined public opinions on scientific issues, this study examined a social issue, immigration, which is very important for the potential consequences that extreme radicalized positions might have on society. Our results show that polarization is not limited to scientific issues but also spreads to social issues, such as immigration. Moreover, in addition to numerical and scientific literacy, already used in previous studies, we have included economic literacy, which we believe is an important competence for evaluating complex issues, such as immigration, for its linkages with individual's political orientation, immigration opposition, and education. Furthermore, the polarization effect of knowledge was investigated in a representative sample of citizens from a southern European city. Southern Europe (Italy, Greece, Spain) is at the forefront of migration flows from Africa, making it a perfect field to study citizens' fears and concerns about immigration.

Many studies have used a measure of political orientation to elicit the knowledge-related polarization effect [25, 42, 78]. While in some cases, this has been a pre-designed choice [e.g., 30], in other cases, such as large-scale representative surveys, it has been an ex-post forced choice due to its availability [e.g., 25]. In deciding the ideological measure to use in our study, we decided to avoid using political orientation. Political orientation is typically elicited by asking respondents to classify themselves on some bipolar dimension, such as republican vs. democrat or liberal vs. conservative. Instead, we preferred to use a measure of cultural worldviews [9]. The reason for this choice was twofold. On the one hand, a standard question about political orientation (right-wing or left-wing) would not adapt well to our context. Indeed, the Italian political landscape is characterized by small and fragmented parties with transversal positions with respect to the standard right-wing or left-wing dichotomy. For example, the 5 Star Movement is a populist party difficult to classify as right or left [88, 89]. It has both right-wing (e.g., anti-immigrant) and left-wing (e.g., guaranteed minimum income) ideologies, as well as both conservative (e.g., NO TAV movement) and liberal (e.g., drug liberalization) ideologies [89].

A second and more important reason is that we believe worldviews are a more detailed and nuanced measure of the underlying ideology of individual beliefs and behaviors than political orientation, with whom they do not fully overlap. Worldviews capture where the individual stands on the spectrum anchored by hierarchical-individualistic beliefs at one pole and egalitarian-communitarian beliefs at the other. They have proved to be successful in identifying a knowledge-related polarizing effect in the case of previous risk perception studies [10]. Moreover, worldview orientations have shown to be significant predictors of an individual's attitudes and behaviors in the face of threats [12, 17, 21], sometimes even more than political orientations [11]. Our study confirms that cultural worldviews may be a valid construct for measuring knowledge-related polarization effects of risk perceptions of social problems.

The results of this study have both theoretical and practical implications. From a theoretical point of view, observing that more knowledgeable and numerically literate people are more influenced by pre-existing ideologies than less knowledgeable people contradicts science communication models that are based on the principle of "information deficit" [40]. Indeed, the polarizing effect highlights a paradoxical aspect of being highly knowledgeable and cognitively able: higher knowledge and ability lead to greater radicalization of public opinions aligning them with the public's cultural worldviews. From a practical point of view, the knowledge-polarization effect contradicts the idea that providing more information to citizens and increasing their knowledge (economic and scientific) or their cognitive ability (numeracy) is a way to reduce social polarization. From a communication and management perspective, these findings are discouraging because they leave little hope for policymakers that more education will be sufficient to align the public's views with those of experts and reduce conflict between experts and the public on issues such as immigration.

Why are more educated people more polarized in their views? A motivated reasoning account has been applied to explain the polarization effects. Motivated reasoning would motivate people to selectively seek, elaborate, and recall information in a way that supports their a priori beliefs [41]. Since highly knowledgeable people also tend to seek more information than less knowledgeable people, the polarization effect would arise from the combination of increased information processing coupled with motivated reasoning [10, 25, 90]. Biased motivated reasoning might reduce scientific and objective messages' effectiveness and intensify opposing positions' crystallization, thus increasing social tension [91]. Some evidence, indeed, suggests that polarization is induced by selective exposure and selective interpretation of information consistent with one's ideology [30]. Motivated reasoning is of particular concern in the context of politicized science [92, 93], but it might be as equally critical in other types of controversial issues related to intolerance [94], such as immigration.

Some research, however, has raised concerns about the robustness of the explanation based on cultural cognition [95]. The fact that higher knowledge is associated with greater cultural polarization of concern about immigration is consistent with the motivated reasoning explanation, yet, other explanations may also apply. For example, numeracy effects may actually be due to variance shared with other types of intelligence, such as verbal ability in solving analogies [30]. It is known, indeed, that more informed individuals are usually more polarized in their political attitudes [e.g., 96] and that multiple forms of intelligence can predict the polarization of political attitudes [97]. This evidence might raise the question of whether our results are due to the specific variables we have examined (i.e., numerical, economic, and science literacy) or, rather, to intelligence. As regards numeracy, it has been shown that individual behavior can be explained by numeracy, even after controlling for intelligence. For example, the positive relationship between numeracy and comprehension of numerical data persists even after controlling for measures of intelligence [98]. Seemingly, conjunction errors are predicted by lower objective numeracy, even after controlling for intelligence measures [99]. Numeracy and intelligence are

certainly strongly related, though they are not perfectly overlapping. On the other hand, recent research on the polarization of COVID-19 risk perception showed that numeracy failed to predict polarization when verbal ability was also measured, suggesting that what might seem an effect of numeracy is indeed an effect of cognitive ability, such as verbal ability [30]. Subsequent studies should measure individual intelligence as well as individual knowledge to directly compare the respective predictive effect and address the question of whether polarized views of immigration are better explained by intelligence rather than knowledge or education.

In the present study, we showed a polarizing effect of objective knowledge (scientific, and economic) and cognitive ability (numeracy) on attitudes toward immigration. The scope of the present study was not to explore the causal mechanisms behind this pattern of results. However, we strongly believe that achieving an understanding of what causes the polarization of beliefs is of theoretical and practical importance. Future studies should investigate these causal explanations, such as group identity or prejudice, and understand their relationship to the cultural polarization of concern about immigration.

## Supporting information

**S1 Table. Reference population.** Reference population (absolute frequencies and percentages) stratified by gender (female/male), age groups (18–35, 36–55, and 56–80 years), and macro-area of residence (South-West, Center-North, North-East).
(DOCX)

**S2 Table. Selected sample.** Selected sample (absolute frequencies and percentages) stratified by gender (female/male), age groups (18–35, 36–55, and 56–80 years), and macro-area of residence (South-West, Center-North, North-East).
(DOCX)

**S3 Table. Response rate.** Response rate (absolute frequencies and percentages) stratified by gender (female/male), age groups (18–35, 36–55 and 56–80 years), and macro-area of residence (South-West, Center-North, North-East).
(DOCX)

**S4 Table. Worldviews items.** Items used in the survey to measure cultural worldviews (10).
(DOCX)

**S5 Table. Worldviews descriptives.** Descriptive statistics for cultural worldviews.
(DOCX)

**S6 Table. Perception of immigration items.** Items used in the survey to measure the perception of immigration.
(DOCX)

**S7 Table. Perception of immigration descriptives.** Descriptive statistics for perception of immigration.
(DOCX)

**S8 Table. Numeracy items.** Items used in the survey to measure numeracy [84].
(DOCX)

**S9 Table. Numeracy descriptives.** Descriptive statistics for numeracy.
(DOCX)

**S10 Table. Economic items.** Items used in the survey to measure economic literacy [46].
(DOCX)

**S11 Table. Economic descriptives.** Descriptive statistics for economic literacy.
(DOCX)

**S12 Table. Science items.** Items used in the survey to measure science literacy [85].
(DOCX)

**S13 Table. Science descriptives.** Descriptive statistics for science literacy.
(DOCX)

**S14 Table. Total descriptives.** Descriptive statistics for total literacy.
(DOCX)

**S15 Table. Correlations.** Pearson correlations coefficients between variables.
(DOCX)

**S1 File. Simulation.** R scripts and explanations of the simulation with the dichotomization
and non-dichotomization of the continuous moderating variable.
(PDF)

## Acknowledgments

We are thankful to the Istituto di Scienze Della Sicurezza (ISSTN) and the Comune di Trento
for logistic support during the data collection. We are thankful to Ellen Peters for helpful com-
ments on the numeracy scale and William Walstad for useful clarifications on the economic
literacy test items.

## Author Contributions

**Conceptualization:** Lucia Savadori, Maria Michela Dickson, Giuseppe Espa.

**Data curation:** Lucia Savadori, Maria Michela Dickson, Giuseppe Espa.

**Formal analysis:** Maria Michela Dickson, Rocco Micciolo, Giuseppe Espa.

**Funding acquisition:** Giuseppe Espa.

**Investigation:** Lucia Savadori, Giuseppe Espa.

**Methodology:** Maria Michela Dickson, Giuseppe Espa.

**Project administration:** Giuseppe Espa.

**Supervision:** Rocco Micciolo.

**Validation:** Rocco Micciolo.

**Writing – original draft:** Lucia Savadori, Maria Michela Dickson, Giuseppe Espa.

**Writing – review & editing:** Lucia Savadori, Maria Michela Dickson, Rocco Micciolo, Giu-
seppe Espa.

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
