## [Decision Letter · Decision Letter 0]

14 Dec 2021

PONE-D-21-32829The polarizing impact of numeracy, economic literacy, and science literacy on the perception of immigrationPLOS ONE

Dear Dr. Savadori,

Thank you for submitting your manuscript to PLOS ONE. After careful consideration, we feel that it has merit but does not fully meet PLOS ONE’s publication criteria as it currently stands. Therefore, we invite you to submit a revised version of the manuscript that addresses the points raised during the review process.

Dear Dr. Savadori,

Thank you for submitting your manuscript “The polarizing impact of numeracy, economic literacy, and science literacy on the perception of immigration” (PONE-D-21-32829) for publication at PLOS ONE. I have now received two reviews from experts in your field of study. In addition, I have read your manuscript independently prior to reading the reviews.

As you can tell from the comments, the reviewers generally see potential in the manuscript but also highlight several problematic aspects which need to be addressed in a possible revision.

Without reiterating every point, the main problems are centered on a lack of theory development and theoretical embedding, data analysis, presentation of results, interpretation of findings, and novelty of the contribution. The reviewers have offered very good, detailed and constructive comments on these issues, and I recommend that you pay close attention to their suggestions.

In my own reading of the manuscript, I found that I largely agree with the reviewers’ comments. I think that the contribution of the manuscript would be significantly sharpened if you included more theory and then interpreted your findings against the backdrop of these theories. The different knowledge predictors (numerical, economic, and scientific literacy) could use better exposition in the introduction, which might help in the development of formal hypotheses as well as in providing more background to the study. R1 suggests several papers which could help with this. In addition to the suggested literature, you might also want to take a look at the following papers on numeracy and prosocial behavior/donations that detail how high vs. low numerate individuals process numerical information regarding people in need: Dickert, Kleber, Peters, & Slovic, 2011, and Kleber, Dickert, Peters, & Florack, 2013. Please note, however, that final decision on this manuscript is not dependent on the inclusion of these two papers (because I have authored them). Similarly, the effects of science and economic literacy on risk judgments could be explained in more detail.  Both reviewers suggest improvements with the data analysis (e.g. on the issue of dichotomization as well as the individual effects of different literacies) as well as with the data presentation (e.g. correlations and comparison of models). I fully share the mentioned concerns and would invite you to address these in detail in the manuscript. Finally, it would be nice to see evidence of critical reflection on the study (e.g. in the form of limitations addressed) and a more careful interpretation of the results. You could also expand on the theoretical as well as practical implications of your study.

Given the comments from the reviewers, the manuscript does not meet the following publication criteria from PLOS ONE: #3 (“Experiments, statistics, and other analyses are performed to a high technical standard and are described in sufficient detail”) and #4 (“Conclusions are presented in an appropriate fashion and are supported by the data”).

I am therefore inviting a major revision. Should you resubmit, please read all comments carefully and respond to each of them in detail.

We look forward to receiving your revised manuscript.

Kind regards,

Stephan Dickert, Ph.D.

Academic Editor

PLOS ONE

Journal Requirements:

2. We note that Figure (1) in your submission contain map images which may be copyrighted. All PLOS content is published under the Creative Commons Attribution License (CC BY 4.0), which means that the manuscript, images, and Supporting Information files will be freely available online, and any third party is permitted to access, download, copy, distribute, and use these materials in any way, even commercially, with proper attribution. For these reasons, we cannot publish previously copyrighted maps or satellite images created using proprietary data, such as Google software (Google Maps, Street View, and Earth). For more information, see our copyright guidelines: http://journals.plos.org/plosone/s/licenses-and-copyright.

1. You may seek permission from the original copyright holder of Figure(s) (1) to publish the content specifically under the CC BY 4.0 license.  

Additional Editor Comments (if provided):

Reviewers' comments:

Reviewer's Responses to Questions

**Comments to the Author**

1. Is the manuscript technically sound, and do the data support the conclusions?

Reviewer #1: Partly

Reviewer #2: Yes

2. Has the statistical analysis been performed appropriately and rigorously? 

Reviewer #1: No

Reviewer #2: Yes

3. Have the authors made all data underlying the findings in their manuscript fully available?

Reviewer #1: Yes

Reviewer #2: No

4. Is the manuscript presented in an intelligible fashion and written in standard English?

Reviewer #1: Yes

Reviewer #2: Yes

5. Review Comments to the Author

Reviewer #1: Major comments

1. Please indicate how duplicate responses were prevented in the online survey (e.g., unique survey links, IP addresses, etc.).

2. Was the survey conducted in Italian or English?

3. Please provide a rationale for choosing the subset of economic literacy questions.

4. Dichotomization of quantitative variables is not ideal and should only be used in very specific, theoretically-derived conditions (MacCallum et al., 2002, citation below). The authors say that this is not an issue because results were similar but don't provide the results for verification. The fact that another article used a non-ideal analysis isn't sufficient justification. Please conduct the analyses using the full ideological scales. I suggest mean-centering your quantitative variables involved in interactions to aid in interpretation of any main effects.

5. "the parameter associated to the non-binary variable (say ® ) resulted however significantly different from zero" is unclear. Please clarify what this means.

6. Table 1 refers to x, y, and z and betas with subscripts. Please indicate which variable each term represents rather than forcing the reader to refer back to the regression equation. I think, but am not sure, that b3 was the interaction, but then page 19 refers to the b3 row for the non-significant effects of worldview. Indicating the effects using words would eliminate the potential for confusion about which coefficient corresponds to which tested effect.

7. Showing polarization with respect to social issues is not novel. Political scientists have long known (e.g., Converse, 2000) that more knowledgeable people are more polarized in their political attitudes. In addition, Ganzach (2018) showed that multiple forms of intelligence predict polarization in political attitudes. It is therefore unclear whether your findings are due to your hypothesized rationale for those variables specifically being related to polarization or whether your findings are due to intelligence, in general.

8. Your discussion that your use of cultural worldviews being necessary because the political landscape in Italy cannot be reduced to unidimensional left-right classification, yet you reduced your measure of ideology to two dimensions and then further conducted a median split. Furthermore, the Kahan measure has two scales that could be analyzed orthogonally. If it is really the case that the worldviews in Italy are more complex, doesn't that argue for analyzing the two scales separately? Please clarify.

9. Your analyses do not support your conclusion that your data "show that cultural worldviews may be an ideal substitute for measuring polarization effects in areas where a two-dimensional measure of political orientation might not be as appropriate." because you reduced your measure to one dimension. Where I think your research could be novel is by showing polarization with the worldview scale in a non-US sample, assuming you do show polarization when worldviews are analyzed as continuous measures. It would be interesting and novel if you were able to show polarization is driven by hierarchical/egalitarian or individualism/communitarian subscales.

10. Very recent research has raised some concerns about the robustness of cultural cognition findings (Persson et al., 2021) or suggested that numeracy effects are due to shared variance with other types of intelligence (Shoots-Reinhard et al., 2021). Given the present approach is strongly dependent on the past findings, some discussion of these other findings seems warranted. These two articles should be addressed in the conclusion at a minimum.

11. Please provide descriptive statistics (e.g., range, mean, standard deviations) for all variables. A correlation table would also be helpful.

here are full references for citations above:

MacCallum, R. C., Zhang, S., Preacher, K. J., & Rucker, D. D. (2002). On the practice of dichotomization of quantitative variables. Psychological Methods, 7(1), 19-40. doi: 10.1037/1082-989x.7.1.19

Converse, P. E. (2000). Assessing the capacity of mass electorates. Annual Review of Political Science, 3(1), 331. doi: 10.1146/annurev.polisci.3.1.331

Ganzach, Y. (2018). Intelligence and the rationality of political preferences. Intelligence,

69, 59–70. https://doi.org/10.1016/j.intell.2018.05.002.

Persson, E., Andersson, D., Koppel, L., Västfjäll, D., & Tinghög, G. (2021). A preregistered replication of motivated numeracy. Cognition, 214, 104768. doi: https://doi.org/10.1016/j.cognition.2021.104768

Shoots-Reinhard, B., Goodwin, R., Bjälkebring, P., Markowitz, D. M., Silverstein, M. C., & Peters, E. (2021). Ability-related political polarization in the COVID-19 pandemic. Intelligence, 88, 101580. doi: https://doi.org/10.1016/j.intell.2021.101580

Minor comments

1. Bruine de Bruin et al. 2020 don't report an interaction in their study; only main effects of ideology and media. However, Shoots-Reinhard et al. do show the funnel pattern for COVID, support for Medicare for all, and for weapons buyback programs. Please cite Shoots-Reinhard or another paper that documented an interaction or remove the COVID mention in the introduction.

2. Missing a citation for definition of financial literacy (page 8). Perhaps Lusardi & Mitchell (2008, American Economic Review) could be used?

3. Please report only two decimal places to make it easier to parse especially tables.

4. What is the purpose of Table 2? Isn't it redundant with Figure 2?

5. The Roccato citation is repeated on page 23.

Lusardi, A., & Mitchell, O. S. (2008). Planning and Financial Literacy: How Do Women Fare? American Economic Review, 98(2), 413-417. doi: 10.1257/aer.98.2.413

Reviewer #2: The study investigated whether individuals’ numerical ability, scientific and economic literacy impact their perception of immigration, taking into account their cultural worldview. The PRI survey used a questionnaire with standardized and ad hoc questions. A representative sample of 551 citizens of a town in the northeast of Italy, thus not a national sample.

I found the paper interesting and I think the practical implications that can be derived are very important. Although it was a fairly smooth read, I think there are aspects to consider to improve its quality.

My comments are as follow:

- I think that the variable Perception of Immigration could have been computed more accurately, considering the weight of every single item. It would be advisable to report Cronbach’s alpha index in addition to proceeding with factor analysis.

- If I understand Table 1 correctly, the models are represented by a column. However, I found Table 1 to be unintuitive and hard to read. I suggest replacing the indices b0, b1, etc with the predictors considered in the 4 models.

In addition, I think it would be useful to do an analysis to indicate which of the 4 tested models is the best (e.g., ANOVA or AIC). Further, I think it would be interesting, also for the practical implications of the study, to have a model that simultaneously estimates the effect of numeracy, economic, and science literacy, unless they have a strong correlation with each other (in which case it should be specified).

Further, it should be made explicit how the Total literacy item was calculated.

- Page 21: Why the mean values of the variable perception of immigration was only estimated for numeracy? I think it should be interesting to consider also economic and science literacy.

- Since you decide to investigate cultural worldview instead of using political orientation, I would suggest discussing it more in deep. For example, are there any studies investigating the correlation between the two measures? This could help support your choice of not considering the political orientation and polarization, even if the Italian political situation is very "confused". I suggest that the link between polarization and cultural worldview is discussed more fully already in the introduction.

- Page 19 line 11: I think is row b2 and not b3. In addition, compared to the results presented, I suggest a more cautious interpretation of the non-significant main effect of the z variable.

Minor suggestions:

- I would not use x, y, and z in the main text, I think it makes the reading not very smooth.

- Figure 2 caption is quite long and quite overly descriptive.

- In the state of the art, I would anticipate (briefly) why the numeracy, economic, and science literacy scales were chosen as variables. I believe that this clarification can be helpful in the reading.

- About science literacy (paragraph 2.4), I would add some references (also from newspapers) to justify the passage on the intertwining between migrants and the increase of viral infections.

- I suggest reviewing the Supplementary Materials, in some of the presented scales references are specified in others not. Additionally, in the Perception of immigration, I believe it would be appropriate to differentiate items from the 1974/2014 General Social Survey (Smith et al., 2017) from those adapted from other work.

- Page 23: There was some problem with the citation of Roccato et al., 2020

- As for the language, I am not a native English speaker, so I will defer to others in evaluating this aspect.

6. PLOS authors have the option to publish the peer review history of their article (what does this mean?). If published, this will include your full peer review and any attached files.

Reviewer #1: No

Reviewer #2: No

---

## [Author Response · Author response to Decision Letter 0]

6 Jul 2022

Dear Stephan Dickert, Academic Editor,

Thank you for the opportunity to revise and resubmit our manuscript. We have gone through the points raised by you and the reviewers and made several changes as a result. Required changes have been highlighted in track changes in the manuscript. Among other things, we would like to draw your attention to the fact that we added an author to the paper (Rocco Micciolo) for the substantive work he provided in responding to Reviewer 1's comment No. 4 (on the issue of dichotomization).

In the remainder of this letter, you will find all issues raised by you and the reviewers, each numbered and followed by our detailed response (in bold). We hope you agree that we have adequately addressed all concerns and improved the manuscript in the process.

Best regards.

RESPONSES TO EDITOR

1. As you can tell from the comments, the reviewers generally see potential in the manuscript but also highlight several problematic aspects which need to be addressed in a possible revision. Without reiterating every point, the main problems are centered on a lack of theory development and theoretical embedding, data analysis, presentation of results, interpretation of findings, and novelty of the contribution. The reviewers have offered very good, detailed and constructive comments on these issues, and I recommend that you pay close attention to their suggestions.

ANSWER: Thank you, we did pay close attention and have revised the manuscript accordingly, as detailed below.

2. In my own reading of the manuscript, I found that I largely agree with the reviewers' comments. I think that the contribution of the manuscript would be significantly sharpened if you included more theory and then interpreted your findings against the backdrop of these theories. The different knowledge predictors (numerical, economic, and scientific literacy) could use better exposition in the introduction, which might help in the development of formal hypotheses as well as in providing more background to the study. R1 suggests several papers which could help with this. In addition to the suggested literature, you might also want to take a look at the following papers on numeracy and prosocial behavior/donations that detail how high vs. low numerate individuals process numerical information regarding people in need: Dickert, Kleber, Peters, & Slovic, 2011, and Kleber, Dickert, Peters, & Florack, 2013. Please note, however, that final decision on this manuscript is not dependent on the inclusion of these two papers (because I have authored them). 

ANSWER: Thank you for noting this lack of theoretical background. We definitely agree with you. In our revised version, we have added more theory on both the models that explain science communication ("deficit model" and "motivated reasoning") as well as more theory on each one of the different predictors (numeracy, economic knowledge, and science knowledge). Having improved the theoretical background led us to significantly sharpen our hypotheses, which are now more clearly stated at the end of each paragraph. In this way, we have included the papers suggested by Reviewer 1 and those related to numeracy regarding people in need.

For example, in the introduction, we added a paragraph on science communication models:

"These studies raised questions about which science communication model was best suited to explain them. On the one hand, the “deficit” model predicts that greater dissemination of scientific knowledge will increase public consensus toward scientific standpoints (e.g., reduce climate change) (Sturgis & Allum, 2004). On the other hand, the evidence shows the opposite, namely that greater knowledge increases the polarization of public opinion toward opposite poles defined by pre-held ideological orientations. A “motivated reasoning” model has been suggested to explain this evidence, suggesting that people filter and process information to support previously held beliefs (Kunda, 1990).”

To introduce the three predictors (numeracy, economic knowledge, and science knowledge), we added a paragraph:

"Of central importance to the knowledge-related polarization effect is the type of knowledge/education/ability considered in the interaction (Shoots-Reinhard et al., 2021). Across studies on polarization, researchers have mostly used education as the knowledge variable to test for the interaction between partnership and opinions, with few exceptions that used science knowledge and others that used cognitive abilities (i.e., numeracy and verbal abilities). The type of knowledge variable chosen has been shown to determine the chance of detecting the interaction with ideology (Shoots-Reinhard et al., 2021). When verbal ability measures were not controlled for, numeracy and ideology did interact to predict outcomes, but they failed to interact when the verbal ability measure was included in the model (Shoots-Reinhard et al., 2021). Thus in the present study, we chose to examine the hypothesized interaction effect (knowledge x ideology), exploring multiple knowledge variables. We used two knowledge predictors (science and economic literacy) and one cognitive ability predictor (numeracy) to broaden the set of knowledge variables that elicit the knowledge-related polarization effect.”

Moreover, we expanded each of the three paragraphs devoted to the three predictors (numerical, economic, and scientific literacy) to include arguments to support our hypotheses. We do not copy and paste them here because they are too long, but you can find them highlighted in tracked changes in the introductory part of the manuscript.

3. Similarly, the effects of science and economic literacy on risk judgments could be explained in more detail. 

ANSWER: Yes, we did so (see point above).

4. Both reviewers suggest improvements with the data analysis (e.g. on the issue of dichotomization as well as the individual effects of different literacies) as well as with the data presentation (e.g. correlations and comparison of models). I fully share the mentioned concerns and would invite you to address these in detail in the manuscript. 

ANSWER: Thank you. We have addressed this issue in answer to the reviews below (see points #4, #9 and #10 Reviewer 1) 

5. Finally, it would be nice to see evidence of critical reflection on the study (e.g. in the form of limitations addressed) and a more careful interpretation of the results. You could also expand on the theoretical as well as practical implications of your study.

ANSWER: We have expanded the conclusions to embrace a more critical evaluation of our results (limitations and discussion of the results) as well as theoretical and practical implications.

" The results of this study have both theoretical and practical implications. From a theoretical point of view, observing that more knowledgeable and numerically literate people are more influenced by pre-existing ideologies than less knowledgeable people contradicts science communication models that are based on the principle of “information deficit” (Sturgis & Allum, 2004). Indeed, the polarizing effect highlights a paradoxical aspect of being highly knowledgeable and cognitively able: higher knowledge and ability lead to greater radicalization of public opinions aligning them with the public’s cultural worldviews. From a practical point of view, the knowledge-polarization effect contradicts the idea that providing more information to citizens and increasing their knowledge (economic and scientific) or their cognitive ability (numeracy) is a way to reduce social conflicts. From a communication and management perspective, these findings are discouraging because they leave little hope for policymakers that more education will be sufficient to align the public's views with those of experts and reduce conflict between experts and the public on issues such as immigration."

And a paragraph related to limitations and future research:

" Some research, however, has raised concerns about the robustness of the explanation based on cultural cognition (Persson et al., 2021). The fact that higher knowledge is associated with greater cultural polarization of concern about immigration is consistent with the motivated reasoning explanation, yet, other explanations may also apply. For example, numeracy effects may actually be due to variance shared with other types of intelligence, such as verbal ability in solving analogies (Shoots-Reinhard et al., 2021). It is known, indeed, that more informed individuals are usually more polarized in their political attitudes (e.g., Converse, 2000) and that multiple forms of intelligence can predict the polarization of political attitudes (Ganzach, 2018). This evidence might raise the question of whether our results are due to the specific variables we have examined (i.e., numerical, economic, and science literacy) or, rather, to intelligence. As regards numeracy, it has been shown that individual behavior can be explained by numeracy, even after controlling for intelligence. For example, the positive relationship between numeracy and comprehension of numerical data persists even after controlling for measures of intelligence (Låg et al., 2014). Seemingly, conjunction errors are predicted by lower objective numeracy, even after controlling for intelligence measures (Liberali et al., 2012). Numeracy and intelligence are certainly strongly related, however they are not perfectly overlapping. On the other side, recent research on the polarization of COVID-19 risk perception showed that numeracy failed to predict polarization when verbal ability was also measured, suggesting that what might seem an effect of numeracy is indeed an effect of congitive ability, such as verbal ability (30). Subsequent studies should measure individual intelligence as well as individual knowledge to directly compare the respective predictive effect and address the question of whether polarized views of immigration are better explained by intelligence rather than knowledge or education.

In the present study, we showed a polarizing effect of objective knowledge (scientific, and economic) and cognitive ability (numeracy) on attitudes toward immigration. The scope of the present study was not to explore the causal mechanisms behind this pattern of results. However, we strongly believe that achieving an understanding of what causes the polarization of beliefs is of theoretical and practical importance. Future studies should investigate these causal explanations, such as group identity or prejudice, and understand their relationship to the cultural polarization of concern about immigration."

RESPONSE TO REVIEWER 1

Dear Reviewer #1, 

we thank you for your careful reading of the manuscript and the useful advices you gave us. We have accepted all of your suggestions except for one point (point #4 concerning the dichotomization of the continuous variable and point #9 and #10). The reasons for our resistance are detailed below. We believe that the manuscript is now much improved with the changes we have made, and we are grateful to you for this.

1. Please indicate how duplicate responses were prevented in the online survey (e.g., unique survey links, I.P. addresses, etc.).

ANSWER: Thank you for noting this. Each participant was sent a unique (and anonymous) code identifier by mail with which they could access the online questionnaire. The code could be used only once. We clarified this in the manuscript adding a couple of sentences as follows:

" Each participant was sent a unique (and anonymous) code identifier by mail with which they could access the online questionnaire. A single usage of the code was allowed. No signed informed consent was collected, but the participant gave electronic informed consent by accessing the questionnaire with their unique code and agreeing to complete it online. The participant needing help in compiling the form contacted the telephone number and provided their unique code to the experimenter, who accessed the questionnaire on their behalf and read the questions to the participant by phone and completed the questionnaire. "

2. Was the survey conducted in Italian or English?

ANSWER: We apologize for the lack of clarity. The survey was conducted in Italian. We have now clarified this point in the method section of the manuscript. We added a sentence:

" The questionnaire was in Italian, the respondents' native language."

3. Please provide a rationale for choosing the subset of economic literacy questions.

ANSWER: Thank you for allowing us to clarify this point. The original Test of Economic Literacy by Walstad, Rebeck, and Butters (2013) includes 45 items covering 20 content standards. For a survey like ours that investigated multiple constructs (5 macro constructs) on large numbers of individuals (a representative sample of the city population), 45 items would have been too many. As there is no short form of the test, we selected a smaller number of 12 items according to two criteria: (i) the items had to be simple enough not to elicit a refusal response from respondents (volunteers) driven by the feeling of being under scrutiny; (ii) the items had to be distributed across different content standards so as to be representative of the different contents of the test. Therefore, we used the U.S. data on item difficulty to select a subset of items that were both easy (over 40% correct responses in the student sample not enrolled in a basic course with economics) and representative of different contents. The 12 selected items had a difficulty ranging from a minimum of 42.3% to a maximum of 66.3% correct responses, with an average of 51% correct responses. In addition, the items investigated the following 9 contents: (a) Economic incentives - prices, wages, profits, etc. (item 8); (b) Voluntary exchange and trade (item 9); (c) Markets and prices (item 13); (d) Supply and demand (items 15 and 17); (e) Money and inflation (items 23 and 25); (f) Interest rates (item 26); (g) Entrepreneurship (item 30); (h) Unemployment and inflation (item 41 and item 42); (i) Fiscal and monetary policy (item 44). 

We have added this information in the manuscript in the section on "Economic Literacy": 

" Items were selected to meet two criteria: (1) they had to be sufficiently easy (more than 40% correct responses in the U.S. student sample not enrolled in a basic course with economics, as reported in (Walstad et al., 2013)); (2) they had to be representative of different test contents. The subset of 12 items selected had a difficulty that ranged from a minimum of 42.3% to a maximum of 66.3% correct responses, with an average of 51% correct responses. In addition, the items investigated the following 9 contents out of 20: (a) Economic incentives - prices, wages, profits, etc (item 8); (b) Voluntary exchange and trade (item 9); (c) Markets and prices (item 13); (d) Supply and demand (items 15 and 17); (e) Money and inflation (items 23 and 25); (f) Interest rates (item 26); (g) Entrepreneurship (item 30); (h) Unemployment and inflation (items 41 and item 42); (i) Fiscal and monetary policy (item 44). Individual score on economic literacy was computed by calculating the number of correct answers (M = 8.91; SD = 2.67).

4. Dichotomization of quantitative variables is not ideal and should only be used in very specific, theoretically-derived conditions (MacCallum et al., 2002, citation below). The authors say that this is not an issue because results were similar but don't provide the results for verification. The fact that another article used a non-ideal analysis isn't sufficient justification. Please conduct the analyses using the full ideological scales. I suggest mean-centering your quantitative variables involved in interactions to aid in interpretation of any main effects.

ANSWER: Thank you for pointing this out. The article by MacCallum et al. (2002) is a very interesting read. The arguments of MacCallum et al. (2002) are certainly right and of great importance. Indeed, the article shows the risks of dichotomization in the presence of continuous reality. And on this assumption, is based. As well as the simulations that are proposed, which start from a continuous reality. Our study's situation is quite different. We strongly believe that our reality is made up of two groups, with a certain degree of overlap. Of course, with the possibility of making classification errors. The scale used in the questionnaire is conceived to identify them. In fact, the items measuring the cultural worldview are aimed at producing a dichotomization. Our paper assumes that there are two pre-existing and predefined subpopulations and that the variable we have used (worldview orientation) helps us to identify them. Following MacCallum et al. (2002), we performed a simulation that demonstrates how using the continuous variable to predict a dichotomized reality reduces the power of the interaction test (which is the parameter of our primary interest). We show some of the most relevant results that emerged from that simulation in the attached file (see S1 File). Because the reader might legitimately have the same concerns as you did, we have added this sentence in the manuscript in the section on data analysis and made available the simulation in supplementary materials. 

" This dichotomization does not constitute a forcing nor an excessive simplification and, especially, does not introduce any distortion in the proposed and estimated models, such as significative interactions which would otherwise not exist. Dichotomization can yield misleading results in the presence of continuous reality (MacCallum et al., 2002). However, we believe that our reality is made up of two groups with a certain degree of overlap. The cultural worldviews scale is conceived to identify them, of course, with the possibility of making classification errors. The worldview orientation scale uses a continuous measurement for research purposes, i.e., the need to elicit truthful answers to sensitive ideological questions. However, the items measuring cultural worldviews are aimed at producing a dichotomization, i.e., a classification of an individual as hierarchical-individualistic or egalitarian-communitarian. Following MacCallum et al. (2002), we performed a simulation that demonstrates how using the continuous variable to predict a dichotomized reality reduces the power of the interaction test (which is the parameter of our primary interest). We show some of the most relevant results that emerged from that simulation in the attached file (see S1 File).

5. "the parameter associated to the non-binary variable (say ® ) resulted however significantly different from zero" is unclear. Please clarify what this means.

ANSWER: Thank you for noticing it. We have deleted this sentence which was definitely not clear. Moreover, it is now unnecessary, given the specifications that we included relative to the dichotomization at the previous point (#4).

6. Table 1 refers to x, y, and z and betas with subscripts. Please indicate which variable each term represents rather than forcing the reader to refer back to the regression equation. I think, but am not sure, that b3 was the interaction, but then page 19 refers to the b3 row for the non-significant effects of worldview. Indicating the effects using words would eliminate the potential for confusion about which coefficient corresponds to which tested effect.

ANSWER: Thank you for the careful reading of the results and Table 1. There was a mistake: b3 should have been b2; we have now corrected it. Moreover, as suggested, we made explicit the variable to which lines b0, b1, b2, and b3 in Table 1 refer: the intercept, the literacy variable, the worldviews, and the interaction, respectively.

7. Showing polarization with respect to social issues is not novel. Political scientists have long known (e.g., Converse, 2000) that more knowledgeable people are more polarized in their political attitudes. In addition, Ganzach (2018) showed that multiple forms of intelligence predict polarization in political attitudes. It is therefore unclear whether your findings are due to your hypothesized rationale for those variables specifically being related to polarization or whether your findings are due to intelligence, in general.

ANSWER: Thank you for suggesting this alternative interpretation of our findings based on the work of Converse (2000) and Ganzach (2018). Both papers fit with our study, so we decided to include them in the discussion to address the issue of whether our findings are due to intelligence and not to the knowledge and ability variables we have measured. The topic is really interesting, and after several considerations, we concluded that it is of course, possible to relate increased knowledge/ability with intelligence. This evidence might raise the question of whether our results are due to the specific variables we have examined (i.e., numerical, economic, and science literacy) or, rather, to intelligence. However, at least for numeracy, there are data that indicate that numeracy explains behavior even when controlling for intelligence. For example, the positive relationship between numeracy and comprehension of numerical data remains even after controlling for measures of intelligence (Låg et al., 2014). Seemingly, lower objective numeracy has been associated with more conjunction errors, even after controlling for intelligence measures (Liberali et al., 2012). Numerical ability and intelligence are therefore related, but they are not completely overlapping. We do not know as regards the other two constructs. Intelligence, therefore, may certainly be part of the explanation, but to rule out the explanation, one should measure intelligence. This issue, indeed, is a perfect ground for further studies. Strongly inspired by your comments, we have added a paragraph about this in the conclusions: 

"Some research, however, has raised concerns about the robustness of the explanation based on cultural cognition (Persson et al., 2021). The fact that higher knowledge is associated with greater cultural polarization of concern about immigration is consistent with the motivated reasoning explanation, yet, other explanations may also apply. For example, numeracy effects may actually be due to variance shared with other types of intelligence, such as verbal ability in solving analogies (Shoots-Reinhard et al., 2021). It is known, indeed, that more informed individuals are usually more polarized in their political attitudes (e.g., Converse, 2000) and that multiple forms of intelligence can predict the polarization of political attitudes (Ganzach, 2018). This evidence might raise the question of whether our results are due to the specific variables we have examined (i.e., numerical, economic, and science literacy) or, rather, to intelligence. As regards numeracy, it has been shown that individual behavior can be explained by numeracy, even after controlling for intelligence. For example, the positive relationship between numeracy and comprehension of numerical data persists even after controlling for measures of intelligence (Låg et al., 2014). Seemingly, conjunction errors are predicted by lower objective numeracy, even after controlling for intelligence measures (Liberali et al., 2012). Numeracy and intelligence are certainly strongly related, though they are not perfectly overlapping. On the other side, recent research on the polarization of COVID-19 risk perception showed that numeracy failed to predict polarization when verbal ability was also measured, suggesting that what might seem an effect of numeracy is indeed an effect of cognitive ability, such as verbal ability (30). Subsequent studies should measure individual intelligence as well as individual knowledge to directly compare the respective predictive effect and address the question of whether polarized views of immigration are better explained by intelligence rather than knowledge or education”.

8. Your discussion that your use of cultural worldviews being necessary because the political landscape in Italy cannot be reduced to unidimensional left-right classification, yet you reduced your measure of ideology to two dimensions and then further conducted a median split.

 ANSWER: You are completely right. We were definitely not very logical in our explanations, as you rightly pointed out, and we are sorry to have caused confusion. We have now clarified that what we meant when we said that the political landscape in Italy is complex and that a two-dimensional measure of political orientation was not appropriate was that the typical measure used for political orientation would not apply to our context. To be more clear, few Italians would be able to tell whether they are more liberal or conservative, there are no bipolar-parties such as republicans and democrats and few would be able to classify themselves as clearly right or left. Perhaps Italy is going through a period of socio-political transformation because we observe that many who used to be left-wingers now vote right-wing not because they have become right-wingers but because they believe that the battles of the left are conducted more by right-wing politicians than by left-wing ones. And vice versa. To a straightforward question, such as "are you right-wing or left-wing" these people would not know what to answer because they are left-wing but vote to the right. We believed instead that a more detailed and nuanced measure of the underlying ideology of individual beliefs and behaviors, such as worldviews, would better serve the purpose of dichotomizing the sample into two polarized groups. We justified this decision in the following way:

" Many studies have used a measure of political orientation to elicit the knowledge-related polarization effect (Drummond & Fischhoff, 2017; Hamilton et al., 2015; Smith et al., 2017). While in some cases this has been a pre-designed choice (e.g., Shoots-Reinhard et al., 2021), in other cases such as large-scale representative surveys, it has been an ex-post forced choice due to its availability (e.g., Drummond & Fischhoff, 2017). In deciding the ideological measure to use in our study we decided to avoid using political orientation. Political orientation is typically elicited by asking respondents to classify themselves on some bipolar dimension, such as, republican vs. democrat or liberal vs. conservative. Instead, we preferred to use a measure of cultural worldviews (Kahan et al., 2011). The reason for this choice was twofold. On the one hand, a standard question about political orientation (right-wing or left-wing) would not adapt well to our context. Indeed, the Italian political landscape is characterized by small and fragmented parties with transversal positions with respect to the standard right-wing or left-wing dichotomy. For example, the 5 Star Movement is a populist party difficult to classify as right or left (Roccato et al., 2020; Verbeek & Zaslove, 2016). It has both right-wing (e.g., anti-immigrant) and left-wing (e.g., guaranteed minimum income) ideologies, as well as both conservative (e.g., NO TAV movement) and liberal (e.g., drug liberalization) ideologies (Roccato et al., 2020). A second and more important reason is that we believe worldviews are a more detailed and nuanced measure of the underlying ideology of individual beliefs and behaviors than political orientation, with whom they do not fully overlap. Worldviews capture where an individual stands on the spectrum anchored by hierarchical-individualistic beliefs at one pole and egalitarian-communitarian beliefs at the other. They have proved to be successful in identifying a knowledge-related polarizing effect in the case of previous risk perception studies (Kahan 2912). Moreover, worldview orientations have been shown to be significant predictors of an individual’s attitudes and behaviors in the face of threats (Chen et al., 2020; Siegrist & Bearth, 2021; Xue et al., 2014), sometimes even more than political orientations (Dryhurst et al., 2020). Our study confirms that cultural worldviews may be a valid construct for measuring knowledge-related polarization effects of risk perceptions of social problems.

9. Furthermore, the Kahan measure has two scales that could be analyzed orthogonally. If it is really the case that the worldviews in Italy are more complex, doesn't that argue for analyzing the two scales separately? Please clarify.

ANSWER: You are correct that Kahan's scale contains two dimensions, each bipolar, whereas we have combined the questions into one dimension (hierarchical-individualistic vs. egalitarian-communitarian). We did not analyze the two scales separately because our purpose was to rank each individual on the basis of a dichotomous dimension, as explained above in point #8. Keeping the two scales separate would have complicated the reading of the results considerably: all results would have been duplicated by two and wouldn't have added much information because the two scales behave quite similarly.

10. Your analyses do not support your conclusion that your data "show that cultural worldviews may be an ideal substitute for measuring polarization effects in areas where a two-dimensional measure of political orientation might not be as appropriate." because you reduced your measure to one dimension. Where I think your research could be novel is by showing polarization with the worldview scale in a non-US sample, assuming you do show polarization when worldviews are analyzed as continuous measures. It would be interesting and novel if you were able to show polarization is driven by hierarchical/egalitarian or individualism/communitarian subscales.

ANSWER: Thank you for the suggestion. Keeping the two scales separate would have complicated the reading of the results considerably: all results would have been duplicated by two and wouldn't have added much information because the two scales behave quite similarly. As mentioned in 8 and 9 we made it clear that our purpose was to assign each individual to a pole of only one dichotomous dimension to show a polarization effect.

11. Very recent research has raised some concerns about the robustness of cultural cognition findings (Persson et al., 2021) or suggested that numeracy effects are due to shared variance with other types of intelligence (Shoots-Reinhard et al., 2021). Given the present approach is strongly dependent on the past findings, some discussion of these other findings seems warranted. These two articles should be addressed in the conclusion at a minimum.

ANSWER: Thank you for directing us to this interesting and very recent publication that we had become aware of but did not manage to include in the first version of the manuscript, which we have done in this revised version. The work by Persson and colleagues is impressive: it is a pre-registered replication of earlier work by Kahan et al 2017 on politically-consistent motivated reasoning. The replication shows that the effect is virtually absent. Although we do not cite Kahan and colleagues 2017 in our manuscript, but it is true that we do refer to "motivated reasoning", pointing to it as a possible explanation for the polarization. In doing so, we aligned ourselves with the literature suggesting this as the explanation. Our personal belief is that motivated reasoning may be one explanation but that there may be other explanations as well. For example, right-wingers might hold value schemas more rigid and unshapeable, thus also resistant to education and acquired knowledge, while left-wingers less so. This might explain why those on the left and right diverge more from each other as their knowledge increases. However, neither our study nor other classic studies of polarization have found conclusive evidence for an explanation of the phenomenon, which remains largely unexplained. The only study that went a bit deep on explanations was the study by Shoots-Reinhard et al., 2021, which suggests that polarization is induced by selective exposure and selective interpretation of information consistent with one's ideology. In the manuscript, we have softened the strength of the explanation calling into question "motivated reasoning", and expanded the discussion a bit, which now reads like this:

"Some research, however, has raised concerns about the robustness of the explanation based on cultural cognition (Persson et al., 2021). The fact that higher knowledge is associated with greater cultural polarization of concern about immigration is consistent with the motivated reasoning explanation, yet, other explanations may also apply. For example, numeracy effects may actually be due to variance shared with other types of intelligence, such as verbal ability in solving analogies (Shoots-Reinhard et al., 2021)..".

12. Please provide descriptive statistics (e.g., range, mean, standard deviations) for all variables. A correlation table would also be helpful.

ANSWER: Thank you, we had forgotten to include this information in the manuscript and supplementary materials. We have now included the summary descriptive statistics (i.e., M and SD) in the method section, in the paragraph devoted to each variable. We also included the extended descriptive statistics (mean, range, standard deviation) for each question and the mean construct in dedicated tables in the supplementary materials (Tables S4-S14). Finally, we made the correlation table (Table S15) available as supplementary material as well. 

Here are full references for citations above:

MacCallum, R. C., Zhang, S., Preacher, K. J., & Rucker, D. D. (2002). On the practice of dichotomization of quantitative variables. Psychological Methods, 7(1), 19-40. doi: 10.1037/1082-989x.7.1.19

Converse, P. E. (2000). Assessing the capacity of mass electorates. Annual Review of Political Science, 3(1), 331. doi: 10.1146/annurev.polisci.3.1.331

Ganzach, Y. (2018). Intelligence and the rationality of political preferences. Intelligence, 69, 59–70. https://doi.org/10.1016/j.intell.2018.05.002.

Persson, E., Andersson, D., Koppel, L., Västfjäll, D., & Tinghög, G. (2021). A preregistered replication of motivated numeracy. Cognition, 214, 104768. doi: https://doi.org/10.1016/j.cognition.2021.104768

Shoots-Reinhard, B., Goodwin, R., Bjälkebring, P., Markowitz, D. M., Silverstein, M. C., & Peters, E. (2021). Ability-related political polarization in the COVID-19 pandemic. Intelligence, 88, 101580. doi: https://doi.org/10.1016/j.intell.2021.101580

Minor comments

13. Bruine de Bruin et al. 2020 don't report an interaction in their study; only main effects of ideology and media. However, Shoots-Reinhard et al. do show the funnel pattern for COVID, support for Medicare for all, and for weapons buyback programs. Please cite Shoots-Reinhard or another paper that documented an interaction or remove the COVID mention in the introduction.

ANSWER: we really appreciate you noting that. Actually, you are right, Bruin de Bruin 2020 talks about polarization but does not look for interaction with education. So we removed the reference and replaced it with the one you suggested, which is much more appropriate. We still kept the Bruin de Bruin 2020 quote but moved it a bit earlier, in the paragraph where we talk about how cultural worldviews explain risk judgments of individuals in the face of threats.

14. Missing a citation for definition of financial literacy (page 8). Perhaps Lusardi & Mitchell (2008, American Economic Review) could be used?

ANSWER: you are absolutely right. We were so focused on economic literacy that we forgot about financial literacy. Yes, Lusardi's work is perfect. We have included it. Thank you very much.

15. Please report only two decimal places to make it easier to parse especially tables.

ANSWER: We agree. We reduced the decimals in the tables and rechecked all the calculations. We realized that we had forgotten to reverse code an item in the immigration perception measure, which we have now done. As a consequence, the decimals in Table 1 have changed slightly.

16. What is the purpose of Table 2? Isn't it redundant with Figure 2?

ANSWER: Yes indeed, we have eliminated Table 2.

17. The Roccato citation is repeated on page 23.

ANSWER: Yes. There was something wrong with my Mendeley, which kept repeating this entry. We have now fixed it. Thank you.

Lusardi, A., & Mitchell, O. S. (2008). Planning and Financial Literacy: How Do Women Fare? American Economic Review, 98(2), 413-417. doi: 10.1257/aer.98.2.413

RESPONSE TO REVIEWER 2

Dear Reviewer #2, 

Thank you for your mindful reading of the manuscript and for your suggestions. We have implemented all of the recommendations you made except for # 2 and #5. Point # 2 was about using the weighted mean and not the arithmetic mean for the dependent variable perception of immigration. This suggestion, which is normally valid, turned out to be unnecessary in this particular case. We have justified our choice in detail and stand ready to modify the analyses if you deem it necessary. As for point #5 we also explained our reasons below, but we are open in case you prefer to include the results in the manuscript or in the Supporting information. All other suggestions have been implemented, and because of this, we think the manuscript is much improved. Thanks.

1. The study investigated whether individuals' numerical ability, scientific and economic literacy impact their perception of immigration, taking into account their cultural worldview. The PRI survey used a questionnaire with standardized and ad hoc questions. A representative sample of 551 citizens of a town in the northeast of Italy, thus not a national sample. I found the paper interesting and I think the practical implications that can be derived are very important. Although it was a fairly smooth read, I think there are aspects to consider to improve its quality.

ANSWER: We're glad you found the work interesting.

2. - I think that the variable Perception of Immigration could have been computed more accurately, considering the weight of every single item. It would be advisable to report Cronbach's alpha index in addition to proceeding with factor analysis.

ANSWER: Sorry for not being able to communicate these things clearly enough in the manuscript. Actually, Cronbach's alpha was calculated (� = .94), but it is now reported in the paper more clearly. Also, the way the dependent variable 'perception of immigration' was constructed has been made clearer. We explained that the immigration perception variable is the simple arithmetic mean (individual by individual) of the scores given by each respondent to 13 opinion questions about immigrants. We clarified that, before performing the calculation of the averages, the scales of some variables were reversed so that higher scores correspond to a more "favorable" position on immigration. More importantly, a PCA was performed as you suggested but we feel it is not necessary to report this analysis in the main manuscript (although we make it available in the SI) and would prefer to use the mean and not the weighted mean, for three reasons: 

(1) the arithmetic mean results in less loss of information and, therefore, greater "recovery" of data; in fact, PCA uses complete data and any partial non-response imposes exclusion from the calculation of the record for the individual who did not answer one or more questions. Below are the results of PCA, which we would prefer not to report in the manuscript, not without first showing the correlation matrices constructed by calculating Sperman's � (if we are to consider the nature of the data, which are to all ranks) and Pearson's linear correlation coefficients r. 

The correlation matrix R calculated with Sperman's �:

 immimp immcult immcrime immiteco immjobs immcit

immimp 1.0000000 0.5772242 0.6368190 0.7268371 0.6024814 0.5971871

immcult 0.5772242 1.0000000 0.5940080 0.5039950 0.6209019 0.5427874

immcrime 0.6368190 0.5940080 1.0000000 0.5832409 0.6305193 0.5013359

immiteco 0.7268371 0.5039950 0.5832409 1.0000000 0.5834355 0.5881605

immjobs 0.6024814 0.6209019 0.6305193 0.5834355 1.0000000 0.5191298

immcit 0.5971871 0.5427874 0.5013359 0.5881605 0.5191298 1.0000000

immrghts 0.5218624 0.4555000 0.4305476 0.4794168 0.4471456 0.5978142

immcosts 0.6826779 0.6153745 0.6486136 0.6664441 0.6713394 0.5789071

immref 0.4902701 0.4315646 0.4222825 0.5064044 0.4882544 0.4987802

immnum 0.7099802 0.6105259 0.6834254 0.6813957 0.6220203 0.6008854

riskperc 0.7259665 0.6371458 0.7201841 0.6574490 0.6508899 0.6146545

riskaffect 0.4808500 0.4023264 0.4213511 0.4814193 0.3855928 0.4001193

riskben 0.7603968 0.5772596 0.6440378 0.7436093 0.6626731 0.6011694

 immrghts immcosts immref immnum riskperc riskaffect

immimp 0.5218624 0.6826779 0.4902701 0.7099802 0.7259665 0.4808500

immcult 0.4555000 0.6153745 0.4315646 0.6105259 0.6371458 0.4023264

immcrime 0.4305476 0.6486136 0.4222825 0.6834254 0.7201841 0.4213511

immiteco 0.4794168 0.6664441 0.5064044 0.6813957 0.6574490 0.4814193

immjobs 0.4471456 0.6713394 0.4882544 0.6220203 0.6508899 0.3855928

immcit 0.5978142 0.5789071 0.4987802 0.6008854 0.6146545 0.4001193

immrghts 1.0000000 0.4730686 0.3918708 0.5148810 0.5643132 0.3273390

immcosts 0.4730686 1.0000000 0.5890304 0.7571954 0.7112919 0.4596726

immref 0.3918708 0.5890304 1.0000000 0.5685539 0.5384318 0.2919514

immnum 0.5148810 0.7571954 0.5685539 1.0000000 0.7365787 0.4700815

riskperc 0.5643132 0.7112919 0.5384318 0.7365787 1.0000000 0.5529421

riskaffect 0.3273390 0.4596726 0.2919514 0.4700815 0.5529421 1.0000000

riskben 0.5619813 0.7192198 0.5029110 0.7575481 0.7387274 0.4925383

 riskben

immimp 0.7603968

immcult 0.5772596

immcrime 0.6440378

immiteco 0.7436093

immjobs 0.6626731

immcit 0.6011694

immrghts 0.5619813

immcosts 0.7192198

immref 0.5029110

immnum 0.7575481

riskperc 0.7387274

riskaffect 0.4925383

riskben 1.0000000

The correlation matrix R constructed by calculating Pearson's linear correlation coefficients r:

 immimp immcult immcrime immiteco immjobs immcit

immimp 1.0000000 0.5997608 0.6138052 0.7455894 0.6060955 0.5878850

immcult 0.5997608 1.0000000 0.5772389 0.5048826 0.5875181 0.5324282

immcrime 0.6138052 0.5772389 1.0000000 0.5604829 0.6078344 0.4759312

immiteco 0.7455894 0.5048826 0.5604829 1.0000000 0.5839685 0.5703349

immjobs 0.6060955 0.5875181 0.6078344 0.5839685 1.0000000 0.4683891

immcit 0.5878850 0.5324282 0.4759312 0.5703349 0.4683891 1.0000000

immrghts 0.5254873 0.4558376 0.4104501 0.4737751 0.4135153 0.5851733

immcosts 0.6828013 0.6137007 0.6481052 0.6593644 0.6518089 0.5528748

immref 0.5058409 0.4006633 0.4036455 0.5120538 0.4515537 0.4636040

immnum 0.7043481 0.6003306 0.6736334 0.6625200 0.5988081 0.5848215

riskperc 0.7306793 0.6395683 0.6815122 0.6757126 0.6190803 0.6207054

riskaffect 0.4968532 0.4258549 0.4206749 0.4995838 0.3913780 0.4284992

riskben 0.7556542 0.5689027 0.6337376 0.7327885 0.6406038 0.5858643

 immrghts immcosts immref immnum riskperc riskaffect

immimp 0.5254873 0.6828013 0.5058409 0.7043481 0.7306793 0.4968532

immcult 0.4558376 0.6137007 0.4006633 0.6003306 0.6395683 0.4258549

immcrime 0.4104501 0.6481052 0.4036455 0.6736334 0.6815122 0.4206749

immiteco 0.4737751 0.6593644 0.5120538 0.6625200 0.6757126 0.4995838

immjobs 0.4135153 0.6518089 0.4515537 0.5988081 0.6190803 0.3913780

immcit 0.5851733 0.5528748 0.4636040 0.5848215 0.6207054 0.4284992

immrghts 1.0000000 0.4522670 0.3732079 0.4948080 0.5536078 0.3433215

immcosts 0.4522670 1.0000000 0.5738581 0.7577632 0.6984142 0.4687585

immref 0.3732079 0.5738581 1.0000000 0.5511433 0.5516489 0.3158073

immnum 0.4948080 0.7577632 0.5511433 1.0000000 0.7137501 0.4759196

riskperc 0.5536078 0.6984142 0.5516489 0.7137501 1.0000000 0.5832320

riskaffect 0.3433215 0.4687585 0.3158073 0.4759196 0.5832320 1.0000000

riskben 0.5455191 0.7087311 0.4995647 0.7448280 0.7265088 0.5085346

 riskben

immimp 0.7556542

immcult 0.5689027

immcrime 0.6337376

immiteco 0.7327885

immjobs 0.6406038

immcit 0.5858643

immrghts 0.5455191

immcosts 0.7087311

immref 0.4995647

immnum 0.7448280

riskperc 0.7265088

riskaffect 0.5085346

riskben 1.0000000

As can be seen, the two matrices are practically superimposable.

Principal component analysis, performed on the 13 variables that collectively define Perception of Immigration, yielded the following results in terms of explained variability:

 [1] 0.60513179 0.05942589 0.05473935 0.05043378 0.04071987 0.03288394

 [7] 0.03177319 0.02850099 0.02541370 0.01954264 0.01817199 0.01711045

[13] 0.01615243

The first principal component explains about 60% of the total variability. The subsequent ones explain a proportion of total variability between 6.4% (the second) and 1.3% (the thirteenth). If one considers it more informative, one can look at the following table. The previous one is calculated, so to speak, "by hand." What follows is the output of R's prcomp function:

Importance of components:

 PC1 PC2 PC3 PC4 PC5 PC6 PC7

Standard deviation 3.3731 1.09968 1.05853 0.97019 0.85919 0.7988 0.79015

Proportion of Variance 0.6009 0.06387 0.05917 0.04971 0.03899 0.0337 0.03297

Cumulative Proportion 0.6009 0.66475 0.72393 0.77364 0.81263 0.8463 0.87930

 PC8 PC9 PC10 PC11 PC12 PC13

Standard deviation 0.73962 0.71282 0.62653 0.57059 0.52089 0.49083

Proportion of Variance 0.02889 0.02683 0.02073 0.01719 0.01433 0.01272

Cumulative Proportion 0.90819 0.93502 0.95575 0.97295 0.98728 1.00000

(2) The weights to be attributed to each individual item (the so-called factor loadings of the PCA) are substantially equal. They assume values from 0.22 to 0.35, as can be seen from the table below:

 immimp immcult immcrime immiteco immjobs immcit immrghts 

 0.2963916 0.2750613 0.2928099 0.2747562 0.2441089 0.2908017 0.2326617 

 immcosts immref immnum riskperc riskaffect riskben 

 0.3514382 0.2210933 0.3281109 0.2740960 0.2384599 0.2548310

(3) The average of the 13 variables we have calculated shows a very high correlation with the score given by the first main component (the so-called scores, i.e. the values of the new variable defined, precisely, by the first main component). This correlation, measured with Pearson's linear correlation coefficient r, is equal to 0.98 approximately. Below we also report the scatter of our dependent variable Y against the scores attributed by the first principal component:

In conclusion, we feel that this evidence supports the choice of using the arithmetic mean of the 13 questions to construct the dependent variable perception of immigration, which is more intuitive and more "informative" than the first main component. Summing, and then averaging, the answers given by the respondents to the 13 questions that contribute to the construction of the variable Y is a good strategy, also because of the "I do not know/do not answer". In the manuscript we have added, in this regard, a clarification in the method section and made the statistics available in the supplementary materials (Tables S6 and S7):

" To be conservative, the 13 items were subjected to a principal component analysis which showed that the principal component explains about 60% of the total variability and that the weights of the items (the factor loadings of the PCA) are substantially equal to each other. In addition, the arithmetic mean of the 13 items has a very high correlation index (r = .98) with the score given by the first principal component (the scores, i.e., the values of the new variable defined, precisely, by the first principal component). We, therefore, considered it more informative to use the simple arithmetic mean as the dependent variable and not a more complex statistic, such as the mean of the items each weighted by the principal component, also because the arithmetic mean entails a lesser loss of information due to missing cases."

3. - If I understand Table 1 correctly, the models are represented by a column. However, I found Table 1 to be unintuitive and hard to read. I suggest replacing the indices b0, b1, etc with the predictors considered in the 4 models.

ANSWER: We did it. The same concern was raised by another reviewer. As suggested, we made explicit the variable to which lines b0, b1, b2, and b3 in Table 1 refer: the intercept, the literacy variable, the worldviews, and the interaction, respectively.

4. In addition, I think it would be useful to do an analysis to indicate which of the 4 tested models is the best (e.g., ANOVA or AIC). 

ANSWER: We included AIC values in the last row of Table 1 and commented on them in the text as follows:

" The last row of Table 1 shows the AIC (Akaike information criterion) values for the four fitted models. The model with the lowest AIC included total literacy as the explanatory variable, while the model with the highest AIC included numeracy as the explanatory variable. The difference between these two AICs was 7.651, indicating, according to Burnham and Anderson (2004), good support for considering the model that included total literacy as an explanatory variable as a better model. A similar result, even if with weaker evidence, was found when considering the AIC difference between this model and those which included economic literacy (3.137) or science literacy (5.658) as the explanatory variable"

5. Further, I think it would be interesting, also for the practical implications of the study, to have a model that simultaneously estimates the effect of numeracy, economic, and science literacy, unless they have a strong correlation with each other (in which case it should be specified).

ANSWER: Yes, the request makes sense. On the one hand, the three variables are quite related:

> x1 <- df$economic

> x2 <- df$numeracy

> x3 <- df$science

> tmp <- cbind(x1,x2,x3)

> round(cor(tmp),3)

 x1 x2 x3

x1 1.000 0.457 0.588

x2 0.457 1.000 0.447

x3 0.588 0.447 1.000

On the other, we tried to fit them all into a model (including the 3 interactions with z -worldviews). The only significant interaction is with x2 (numeracy); the variable x3 (science) is not significant (when the other two are included), while x1 (economic) is significant as a main effect. 

(note, only significant interactions are shown here, except for the variable included in the interaction)

 Estimate Std. Error t value Pr(>|t|) 

(Intercept) 2.94944 0.19247 15.324 < 2e-16 ***

z -0.15534 0.20395 -0.762 0.44658 

x2 0.09704 0.05095 1.904 0.05738 . 

x1 0.03143 0.01210 2.597 0.00966 **

z:x2 -0.15095 0.05874 -2.570 0.01044 *

Results show that: 

When z = 0 we have that the expected values are given by 2.95 + 0.031xeconomic + 0.097xnumeracy

When z = 1 we have that the expected values are given by (2.95-0.16) + 0.031xeconomic + (0.097-0.15)x numeracy. Thus, with economic being equal, as numeracy increases, the predicted values in the group with z=0 grow, while in the group with z=1, again with economic being equal, as numeracy increases, the predicted values decrease (or rather, remain essentially constant).

In both groups (z=0 and z=1), with equal numeracy, as economic increases, the expected values increase (in the same way in both groups). In both groups (z=0 and z=1), at equal numeracy and economic, the predicted values do not depend on science.

Given these results, we believe it is more elegant to show a separate model for each predictor. Indeed, the purpose of our work was not to have a competition among predictors to see which among the three predictors was more powerful in determining the effect but rather to observe how each determined the effect. The three predictors, in our view, measure related but also very different, constructs, otherwise, we would not have studied all three of them. In the introductory part of the paper we expand on each of them by noting how each has its own distinctive features in relation to the immigration issue. In conclusion, we believe the proposed solution with the three predictors examined in three separate models is better. However, if the editor prefers, we can easily edit or include also this analysis in the manuscript.

6. Further, it should be made explicit how the Total literacy item was calculated.

ANSWER: Sorry if we weren't clear enough. The total literacy variable was calculated as the mean of the aggregated knowledge measures. We specified this in the methods section.

7. - Page 21: Why the mean values of the variable perception of immigration was only estimated for numeracy? I think it should be interesting to consider also economic and science literacy.

ANSWER: Reviewer 1 says that Table 2 basically replicates the data already presented in Figure 2 and says to remove it. You tell us to add the other data for science, and economic. We decided that it was better to remove it because it does not tell something different than Figure 2.

8. - Since you decide to investigate cultural worldview instead of using political orientation, I would suggest discussing it more in deep. For example, are there any studies investigating the correlation between the two measures? This could help support your choice of not considering the political orientation and polarization, even if the Italian political situation is very "confused". I suggest that the link between polarization and cultural worldview is discussed more fully already in the introduction.

ANSWER: Sorry for the confusion. We did write that we decided to investigate cultural worldview instead of using political orientation but we did not mean it, literally. To be more clear, we choose to measure worldviews not as a "substitute" or "proxy" of political worldviews but because we believed that they were better suited to measure knowledge-related polarization, than political orientation, when dealing with attitudes and behaviors in front of threats (such as immigration). As you suggested we now have detailed our arguments in the paper, both in the introduction and in the conclusion sections.

In the introduction:

Most studies on the polarization of beliefs used political orientation as the polarizing variable. Instead, following Kahan et al. (2012), we used a more nuanced underlying ideological measure, i.e., worldviews, to capture where the individual lies on the ideological spectrum represented by hierarchical-individualistic views on one side and egalitarian-communitarian views on the other. Individual preferences on social problems such as gun control, nuclear waste disposals, COVID-19, and climate change are strongly influenced by cultural worldviews (Cherry et al., 2017; Dryhurst et al., 2020; Kahan et al., 2011; Siegrist & Bearth, 2021). Indeed, in a study of 6,991 individuals across the world, an individualistic worldview predicted COVID-19-related attitudes and behaviors more than all other variables (including political orientation) in five out of the ten countries surveyed (UK, Germany, Sweden, Spain, and Japan) (Dryhurst et al., 2020). Moreover, worldviews have been successfully used in a prior study on risk perception and the polarizing impact of knowledge (Kahan, 2012). We, therefore, measured individual worldviews to categorize individuals into hierarchical-individualistic vs. egalitarian-communitarian and measure the polarizing impact that this underlying ideology induces when put in interaction with personal knowledge..

In the conclusions:

" Many studies have used a measure of political orientation to elicit the knowledge-related polarization effect (Drummond & Fischhoff, 2017; Hamilton et al., 2015; Smith et al., 2017). While in some cases this has been a pre-designed choice (e.g., Shoots-Reinhard et al., 2021), in other cases such as large-scale representative surveys, it has been an ex-post forced choice due to its availability (e.g., Drummond & Fischhoff, 2017). In deciding the ideological measure to use in our study we decided to avoid using political orientation. Political orientation is typically elicited by asking respondents to classify themselves on some bipolar dimension, such as, republican vs. democrat or liberal vs. conservative. Instead, we preferred to use a measure of cultural worldviews (Kahan et al., 2011). The reason for this choice was twofold. On the one hand, a standard question about political orientation (right-wing or left-wing) would not adapt well to our context. Indeed, the Italian political landscape is characterized by small and fragmented parties with transversal positions with respect to the standard right-wing or left-wing dichotomy. For example, the 5 Star Movement is a populist party difficult to classify as right or left (Roccato et al., 2020; Verbeek & Zaslove, 2016). It has both right-wing (e.g., anti-immigrant) and left-wing (e.g., guaranteed minimum income) ideologies, as well as both conservative (e.g., NO TAV movement) and liberal (e.g., drug liberalization) ideologies (Roccato et al., 2020). A second and more important reason is that we believe worldviews are a more detailed and nuanced measure of the underlying ideology of individual beliefs and behaviors than political orientation, with whom they do not fully overlap. Worldviews capture where an individual stands on the spectrum anchored by hierarchical-individualistic beliefs at one pole and egalitarian-communitarian beliefs at the other. They have proved to be successful in identifying a knowledge-related polarizing effect in the case of previous risk perception studies (Kahan 2912). Moreover, worldview orientations have been shown to be significant predictors of an individual’s attitudes and behaviors in the face of threats (Chen et al., 2020; Siegrist & Bearth, 2021; Xue et al., 2014), sometimes even more than political orientations (Dryhurst et al., 2020). Our study confirms that cultural worldviews may be a valid construct for measuring knowledge-related polarization effects of risk perceptions of social problems

9. - Page 19 line 11: I think is row b2 and not b3. In addition, compared to the results presented, I suggest a more cautious interpretation of the non-significant main effect of the z variable.

ANSWER: Yes thank you, you are right, here was a mistake: b3 should have been b2; we have now corrected it.

10. - I would not use x, y, and z in the main text, I think it makes the reading not very smooth.

ANSWER: Thank you for the advice. Indeed, you are right. We have deleted references to x, y and z in all sections of the manuscript, as well in the figure and table captions, except for the section on "data analysis".

11. - Figure 2 caption is quite long and quite overly descriptive.

ANSWER: We reduced the length of the caption sensibly by eliminating the reference to x, z and y, shortening the text, but if we need to shorten it further, we can do it by moving part of the caption into the main text if necessary. Please just ask us. 

12. - In the state of the art, I would anticipate (briefly) why the numeracy, economic, and science literacy scales were chosen as variables. I believe that this clarification can be helpful in the reading.

ANSWER: This was an excellent advise, thank you. In the State of the Art we have now included a paragraph which anticipates the decision to measure the three predictors (science, economic and number literacy). The paragraph is the following: 

" Of central importance to the knowledge-related polarization effect is the type of knowledge/education/ability considered in the interaction (Shoots-Reinhard et al., 2021). Across studies on polarization, researchers have mostly used education as the knowledge variable to test for the interaction between partnership and opinions, with few exceptions that used science knowledge and others that used cognitive abilities (i.e., numeracy and verbal abilities). The type of knowledge variable chosen has been shown to determine the chance of detecting the interaction with ideology (Shoots-Reinhard et al., 2021). When verbal ability measures were not controlled for, numeracy and ideology did interact to predict outcomes, but they failed to interact when the verbal ability measure was included in the model (Shoots-Reinhard et al., 2021). Thus in the present study, we chose to examine the hypothesized interaction effect (knowledge x ideology), exploring multiple knowledge variables. We used two knowledge predictors (science and economic literacy) and one cognitive ability predictor (numeracy) to broaden the set of knowledge variables that elicit the knowledge-related polarization effect. 

."

13. - About science literacy (paragraph 2.4), I would add some references (also from newspapers) to justify the passage on the intertwining between migrants and the increase of viral infections.

ANSWER: You're right, we make a lot of statements but don't support them with adequate references. We have now added references. We believe the paragraph is much improved now:

" Science literacy is the knowledge of basic scientific facts (National Science Board & National Science Foundation, 2020). While scientific knowledge is usually not a significant predictor of risk perception per se (Ho et al., 2019; Kahan et al., 2012), it is a significant factor in polarizing public opinions about climate change (Drummond & Fischhoff, 2017; Kahan et al., 2012). Individuals with higher science literacy showed the greatest cultural-worldviews polarization for climate change risks (Kahan et al., 2012). Seemingly, individuals with greater science knowledge showed more political polarization on issues such as stem cell research, the big bang, human evolution, and climate change (Drummond & Fischhoff, 2017). On a similar line, greater attention to scientific news increased support for policies aimed at reducing climate change for strong liberals but reduced support for strong conservatives (Hart et al., 2015). Science knowledge and fear of immigration share a common ground when it comes to viruses and diseases. Indeed, people might fear immigrants thinking that they can be vehicles for viruses and diseases. Concerns about immigrants and disease have been constantly registered throughout history. For example, Merkel and Stern (2002) explored why in three periods, from 1880 to the present, immigrants have been stigmatized as the etiology of a variety of diseases, despite the data do not support such a narrative. Human mobility, indeed, was historically associated with the spread of infectious diseases (Castelli & Sulis, 2017), however this relationship no longer existing in the contemporary age. Despite this, fear is still supported by the media who portray immigrants as disease spreaders (Esses et al., 2013). For example, during the of COVID-19 outbreak in Italy, the question of whether more immigrants should be brought into the country from the sea borders was also deeply intertwined with the threat they posed as positive drivers of viral infection (Rowe et al., 2021). However, this might be especially true for individuals opposing immigration who historically hold a more ideological attitude of right-wing authoritarianism. We, therefore, predicted that greater science knowledge would interact with cultural worldview orientations in explaining public opinion on immigration. Thus, we anticipated those people with greater science knowledge, and egalitarian-communitarian orientation might show more extreme positive opinions on immigration, while those people with greater science knowledge and hierarchical-individualistic orientation might show more strong negative opinions on immigration. “

14. - I suggest reviewing the Supplementary Materials, in some of the presented scales references are specified in others not. Additionally, in the Perception of immigration, I believe it would be appropriate to differentiate items from the 1974/2014 General Social Survey (Smith et al., 2017) from those adapted from other work.

ANSWER: The supplementary materials have been enriched with new tables and analyses. We have standardized the references and we have also modified, as you suggested, the table of items for measuring perceptions of immigration by inserting an asterisk to indicate those items that were taken from the GSS. Thanks for the suggestions and thanks for your careful reading.

15. - Page 23: There was some problem with the citation of Roccato et al., 2020

ANSWER: Yes, thank you we fixed it. Mendley did something weird with this entry.

16. - As for the language, I am not a native English speaker, so I will defer to others in evaluating this aspect.

ANSWER: We, too, are not native English speakers. We used proofreading software to double-check the wording, but if required, we may have the paper reviewed by a private proofreading agency.

---

## [Decision Letter · Decision Letter 1]

2 Sep 2022

The polarizing impact of numeracy, economic literacy, and science literacy on the perception of immigration

PONE-D-21-32829R1

Dear Dr. Savadori,

We’re pleased to inform you that your manuscript has been judged scientifically suitable for publication and will be formally accepted for publication once it meets all outstanding technical requirements.

Kind regards,

Stephan Dickert, Ph.D.

Academic Editor

PLOS ONE

Additional Editor Comments (optional):

Reviewers' comments:

Reviewer's Responses to Questions

**Comments to the Author**

1. If the authors have adequately addressed your comments raised in a previous round of review and you feel that this manuscript is now acceptable for publication, you may indicate that here to bypass the “Comments to the Author” section, enter your conflict of interest statement in the “Confidential to Editor” section, and submit your "Accept" recommendation.

Reviewer #1: All comments have been addressed

Reviewer #2: All comments have been addressed

2. Is the manuscript technically sound, and do the data support the conclusions?

Reviewer #1: Yes

Reviewer #2: Yes

3. Has the statistical analysis been performed appropriately and rigorously? 

Reviewer #1: Yes

Reviewer #2: Yes

4. Have the authors made all data underlying the findings in their manuscript fully available?

Reviewer #1: Yes

Reviewer #2: No

5. Is the manuscript presented in an intelligible fashion and written in standard English?

Reviewer #1: Yes

Reviewer #2: Yes

6. Review Comments to the Author

Reviewer #1: I found this manuscript to be much improved. I especially appreciated the thoughtful and convincing response to the dichotomization issue--the extra explanation and analyses were extremely helpful. It seems possible that Kahan's cultural worldview measure might be more predictive cross-culturally than liberal/conservative scales that are so prevalent in the U.S. and could be another factor in predicting when and for whom people higher in intelligence/knowledge/literacy are more polarized. Predicting that polarization is predicated on being able to measure the variable motivating the polarization in the first place. I think that if you wanted to, you could study this in future research. It may be another factor explaining why some researchers have not found polarization--perhaps they are measuring the wrong worldview (i.e., liberal/conservative vs. egalitarian/hierarchical).

Reviewer #2: I thank the authors for resolving the critical issues reported during the first review.

I report some minor typos.

- In the abstarct (line 33): it's not clear what skill you're talking about.

- line 265: “i.e. immigration” should be corrected in “i.e., immigration”

- line 301: “For example, during of COVID-19 outbreak in Italy” should be corrected in ”For example, during COVID-19 outbreak in Italy”

- line 347 and 377: Please correct the fonts for R and the package used

- line 734: in “scientific, and economic” please remove the comma

7. PLOS authors have the option to publish the peer review history of their article (what does this mean?). If published, this will include your full peer review and any attached files.

Reviewer #1: No

Reviewer #2: **Yes: **Marta Caserotti

---

## [Editor Report · Acceptance letter]

28 Sep 2022

PONE-D-21-32829R1 

The polarizing impact of numeracy, economic literacy, and science literacy on the perception of immigration 

Dear Dr. Savadori:

I'm pleased to inform you that your manuscript has been deemed suitable for publication in PLOS ONE. Congratulations! Your manuscript is now with our production department. 

Kind regards, 

on behalf of

Dr. Stephan Dickert 

Academic Editor

PLOS ONE